# Deep Learning with Label Differential Privacy

**Badih Ghazi**
Google Research
badihghazi@google.com

**Noah Golowich**[*]
EECS, MIT
nzg@mit.edu

**Ravi Kumar**
Google Research
ravi.k53@gmail.com

**Pasin Manurangsi**
Google Research
pasin@google.com

**Chiyuan Zhang**
Google Research
chiyuan@google.com

## Abstract

The Randomized Response (RR) algorithm [96] is a classical technique to improve robustness in survey aggregation, and has been widely adopted in applications with differential privacy guarantees. We propose a novel algorithm, *Randomized Response with Prior* (RRWithPrior), which can provide more accurate results while maintaining the same level of privacy guaranteed by RR. We then apply RRWithPrior to learn neural networks with *label* differential privacy (LabelDP), and show that when only the label needs to be protected, the model performance can be significantly improved over the previous state-of-the-art private baselines. Moreover, we study different ways to obtain priors, which when used with RRWithPrior can additionally improve the model performance, further reducing the accuracy gap between private and non-private models. We complement the empirical results with theoretical analysis showing that LabelDP is provably easier than protecting both the inputs and labels.

## 1 Introduction

The widespread adoption of machine learning in recent years has increased the concerns about the privacy of individuals whose data is used during the model training. Differential privacy (DP) [34, 33] has emerged as a popular privacy notion that has been the basis of several practical deployments in industry [35, 82, 43, 9, 29] and the U.S. Census [3].

A classical algorithm—that predates DP and was initially designed to eliminate evasive answer biases in survey aggregation—is *Randomized Response* (RR) [96]: when the input is an element from a finite alphabet, the output is equal to the input with a certain probability, and is a uniform random other element from the alphabet with the remaining probability. This simple algorithm is shown to satisfy the strong notion of *local* DP [38, 58], whereby the response of each user is protected, in contrast with the so-called *central* DP setting where a curator has access to the raw user data, and only the output of the curator is required to be DP. We note that schemes building on RR have been studied in several previous works on DP estimation (e.g., [30, 57]), and have been deployed in practice [82].

Meanwhile, the large error incurred by RR (e.g., [17]) has stimulated significant research aiming to improve its accuracy, mostly by relaxing to weaker privacy models (e.g., [36, 24, 4]). In this work, we use a different approach and seek to improve RR by leveraging available *prior* information. (A recent work of Liu et al. [63] also used priors to improve accuracy, but in the context of the DP *multiplicative weights* algorithm, which applies in the *central* DP setting.) The prior information can consist of domain-specific knowledge, (models trained on) publicly available data, or historical

---

[*]Part of this work was done while at Google Research.

runs of a training algorithm. Our algorithm is presented in Section 3 (Algorithm 2). At a high level, given a prior distribution $\mathbf{p}$ on the alphabet, the algorithm uses $\mathbf{p}$ to prune the alphabet. If the prior is reliable and even if the alphabet is only minimally pruned, the probability that the output equals the input is larger than when RR is applied to the entire alphabet. On the other hand, if the prior is uniform over the entire alphabet, then our algorithm recovers the classical RR. To implement the above recipe, one needs to specify how to effect the pruning using $\mathbf{p}$. It turns out that the magnitude of pruning can itself vary depending on $\mathbf{p}$, but we can obtain a closed-form formula for determining this. Interestingly, by studying a suitable linear program, we show that the resulting RRWithPrior strategy is *optimal* in that among all $\varepsilon$-DP algorithms, it maximizes the probability that the output equals the input when the latter is sampled from $\mathbf{p}$ (Theorem 3).

## 1.1 Applications to Learning with Label Differential Privacy

There have been a great number of papers over the last decade that developed DP machine learning algorithms (e.g., [19, 102, 88, 83–85, 78]). In the case of deep learning, the seminal work of Abadi et al. [2] introduced a DP training framework (DP-SGD) that was integrated into TensorFlow [80] and PyTorch [92]. Despite numerous followup works, including, e.g., [75–77, 67, 98, 71, 20, 93], and extensive efforts, the accuracy of models trained with DP-SGD remains significantly lower than that of models trained without DP constraints. Notably, for the widely considered CIFAR-10 dataset, the highest reported accuracy for DP models is $69.3\%$ [93], which strikingly relies on *handcrafted* visual features despite that in non-private scenarios *learned* features long been shown to be superior. Even using pre-training with external (CIFAR-100) data, the best reported DP accuracy, $73\%$[2], is still far below the non-private baselines ($> 95\%$). The performance gap becomes a roadblocker for many real-world applications to adopt DP. In this paper, we focus on a more restricted, but important, special case where the DP guarantee is only required to hold with respect to the labels, as described next.

In the *label differential privacy* (LabelDP) setting, the *labels* are considered sensitive, and their privacy needs to be protected, while the input points are not sensitive. This notion has been studied in the PAC setting [18, 13] and for the particular case of sparse linear regression [94], and it captures several practical scenarios. Examples include: (i) computational advertising where the impressions are known to the Ad Tech[3], and thus considered non-sensitive, while the conversions reveal user interest and are thus private (see, e.g., Nalpas and Dutton [70] and [8]), (ii) recommendation systems where the choices are known, e.g., to the streaming service provider, but the user ratings are considered sensitive, and (iii) user surveys and analytics where demographic information (e.g., age, gender) is non-sensitive but income is sensitive—in fact, this was the motivating reason for Warner [96] to propose RR many decades ago! We present a novel *multi-stage algorithm* (LP-MST) for training deep neural networks with LabelDP that builds on top of RRWithPrior (see Section 3 and Algorithm 3), and we benchmark its empirical performance (Section 5) on multiple datasets, domains, and architectures, including the following.

- On CIFAR-10, we show that it achieves $20\%$ higher accuracy than DP-SGD[4].
- On the more challenging CIFAR-100, we present the first non-trivial DP learning results.
- On MovieLens, which consists of user ratings of movies, we show improvements via LP-MST.

In some applications, domain specific algorithms can be used to obtain priors directly without going through multi-stage training. For image classification problems, we demonstrate how priors computed from a (non-private) *self-supervised learning* [22, 23, 44, 53, 16] phase on the input images can be used to achieve higher accuracy with a LabelDP guarantee with extremely small privacy budgets ($\varepsilon \leq 0.1$, see Section 5.2 for details).

We note that due to the requirement of DP-SGD to compute and clip *per-instance gradient*, it remains technically challenging to scale to larger models or mini-batch sizes, despite numerous attempts to minigate this problem [42, 5, 26, 90]. On the other hand, our formulation allows us to use state-of-

---

[2]For DP parameters of $\varepsilon = 8$ and $\delta = 10^{-5}$, cited from Abadi et al. [2, Figure 6]. For a formal definition of DP, we refer the reader to Definition 2.1.

[3]Ad tech (abbreviating Advertising Technology) comprises the tools that help agencies and brands target, deliver, and analyze their digital advertising efforts; see, e.g., `blog.hubspot.com/marketing/what-is-ad-tech`.

[4]We remark that the notion of $\varepsilon$-DP in [2, 76, 75] is *not* directly comparable to $\varepsilon$-Label DP in our work in that they use the addition/removal notion whereas we use the substitution one. Please see the Supplementary Material for more discussion on this.

the-art deep learning architectures such as ResNet [51]. We also stress that our LP-MST algorithm goes beyond deep learning methods that are robust to label noise. (See [87] for a survey of the latter.)

Our empirical results suggest that protecting the privacy of labels can be significantly easier than protecting the privacy of both inputs and labels. We find further evidence to this by showing that for the special case of stochastic convex optimization (SCO), the sample complexity of algorithms privatizing the labels is much smaller than that of algorithms privatizing both labels and inputs; specifically, we achieve *dimension-independent* bounds for LabelDP (Section 6). We also show that a good prior can ensure smaller population error for non-convex loss. (Details are in the Supplementary Material.)

## 2  Preliminaries

For any positive integer $K$, let $[K] := \{1, \ldots, K\}$. *Randomized response* (RR) [96] is the following: let $\varepsilon \geq 0$ be a parameter and let $y \in [K]$ be the true value known to $\mathsf{RR}_\varepsilon$. When an *observer* queries the value of $y$, $\mathsf{RR}_\varepsilon$ responds with a random draw $\tilde{y}$ from the following probability distribution:

$$\Pr[\tilde{y} = \hat{y}] = \begin{cases} \frac{e^\varepsilon}{e^\varepsilon + K - 1} & \text{for } \hat{y} = y, \\ \frac{1}{e^\varepsilon + K - 1} & \text{otherwise.} \end{cases} \tag{1}$$

In this paper, we focus on the application of learning with label differential privacy. We recall the definition of differential privacy (DP), which is applicable to any notion of *neighboring datasets*. For a textbook reference, we refer the reader to Dwork and Roth [32].

**Definition 2.1** (Differential Privacy (DP) [33, 34])**.** Let $\varepsilon, \delta \in \mathbb{R}_{\geq 0}$. A randomized algorithm A taking as input a dataset is said to be $(\varepsilon, \delta)$-*differentially private* ($(\varepsilon, \delta)$-DP) if for any two *neighboring datasets* $\mathbf{D}$ and $\mathbf{D}'$, and for any subset $S$ of outputs of A, it is the case that $\Pr[\mathsf{A}(\mathbf{D}) \in S] \leq e^\varepsilon \cdot \Pr[\mathsf{A}(\mathbf{D}') \in S] + \delta$. If $\delta = 0$, then A is said to be $\varepsilon$-*differentially private* ($\varepsilon$-DP).

When applied to machine learning methods in general and deep learning in particular, DP is usually enforced on the weights of the trained model [see, e.g., 19, 59, 2]. In this work, we focus on the notion of label differential privacy.

**Definition 2.2** (Label Differential Privacy)**.** Let $\varepsilon, \delta \in \mathbb{R}_{\geq 0}$. A randomized training algorithm A taking as input a dataset is said to be $(\varepsilon, \delta)$-*label differentially private* ($(\varepsilon, \delta)$-LabelDP) if for any two training datasets $\mathbf{D}$ and $\mathbf{D}'$ that differ in the *label* of a *single example*, and for any subset $S$ of outputs of A, it is the case that $\Pr[\mathsf{A}(\mathbf{D}) \in S] \leq e^\varepsilon \cdot \Pr[\mathsf{A}(\mathbf{D}') \in S] + \delta$. If $\delta = 0$, then A is said to be $\varepsilon$-*label differentially private* ($\varepsilon$-LabelDP).

All proofs skipped in the main body are given in the Supplementary Material.

## 3  Randomized Response with Prior

In many real world applications, a prior distribution about the labels could be publicly obtained from domain knowledge and help the learning process. In particular, we consider a setting where for each (private) label $y$ in the training set, there is an associated prior $\mathbf{p} = (p_1, \ldots, p_K)$. The goal is to output a randomized label $\tilde{y}$ that maximizes the probability that the output is correct (or equivalently maximizes the signal-to-noise ratio), i.e., $\Pr[y = \tilde{y}]$. The privacy constraint here is that the algorithm should be $\varepsilon$-DP with respect to $y$. (It need *not* be private with respect to the prior $\mathbf{p}$.)

We first describe our algorithm RRWithPrior by assuming access to such priors.

### 3.1  Algorithm: RRWithPrior

We build our RRWithPrior algorithm with a subroutine called RRTop-$k$, as shown in Algorithm 1, which is a modification of randomized response where we only consider the set of $k$ labels $i$ with largest $p_i$. Then, if the input label $y$ belongs to this set, we use standard randomized response on this set. Otherwise, we output a label from this set uniformly at random.

The main idea behind RRWithPrior is to dynamically estimate an optimal $k^*$ based on the prior $\mathbf{p}$, and run RRTop-$k$ with $k^*$. Specifically, we choose $k^*$ by maximizing $\Pr[\mathsf{RRTop}\text{-}k(y) = y]$. It is not

---

**Algorithm 1** `RRTop-`$k$

---

**Input:** A label $y \in [K]$

**Parameters:** $k \in [K]$, prior $\mathbf{p} = (p_1, \ldots, p_K)$

1. Let $Y_k$ be the set of $k$ labels with maximum prior probability (with ties broken arbritrarily).
2. If $y \in Y_k$, then output $y$ with probability $\frac{e^\varepsilon}{e^\varepsilon + k - 1}$ and output $y' \in Y_k \setminus \{y\}$ with probability $\frac{1}{e^\varepsilon + k - 1}$.
3. If $y \notin Y_k$, output an element from $Y_k$ uniformly at random.

---

hard to see that this expression is exactly equal to $\frac{e^\varepsilon}{e^\varepsilon + k - 1} \cdot \left( \sum_{\tilde{y} \in Y_k} p_{\tilde{y}} \right)$ if $y \sim \mathbf{p}$. `RRWithPrior` is presented in Algorithm 2.

---

**Algorithm 2** `RRWithPrior`

---

**Input:** A label $y \in [K]$

**Parameters:** prior $\mathbf{p} = (p_1, \ldots, p_K)$

1. For $k \in [K]$:
    (a) Compute $w_k := \frac{e^\varepsilon}{e^\varepsilon + k - 1} \cdot \left( \sum_{\tilde{y} \in Y_k} p_{\tilde{y}} \right)$, where $Y_k$ is the set of $k$ labels with maximum prior probability (ties broken arbitrarily).
2. Let $k^* = \arg\max_{k \in [K]} w_k$.
3. Return an output of `RRTop-`$k$ $(y)$ with $k = k^*$.

---

### 3.1.1 Privacy Analysis

It is not hard to show that `RRTop-`$k$ is $\varepsilon$-DP.

**Lemma 1.** `RRTop-`$k$ *is* $\varepsilon$-*DP.*

The privacy guarantee of `RRWithPrior` follows immediately from that of `RRTop-`$k$ (Lemma 1) since our choice of $k$ does not depend on the label $y$:

**Corollary 2.** `RRWithPrior` *is* $\varepsilon$-*DP.*

For learning with a `LabelDP` guarantee, we first use `RRWithPrior` to query a randomized label for each example of the training set, and then apply a general learning algorithm that is robust to random label noise to this dataset. Note that unlike `DP-SGD` [2] that makes new queries on the gradients in every training epoch, we query the randomized label *once* and reuse it in all the training epochs.

### 3.2 Optimality of `RRWithPrior`

In this section we will prove the optimality of `RRWithPrior`. For this, we will need additional notation. For any algorithm R that takes as input a label $y$ and outputs a randomized label $\tilde{y}$, we let $\mathrm{Obj}_{\mathbf{p}}(\mathsf{R})$ denote the probability that the output label is equal to the input label $y$ when $y$ is distributed as $\mathbf{p}$; i.e., $\mathrm{Obj}_{\mathbf{p}}(\mathsf{R}) = \Pr_{y \sim \mathbf{p}}[\mathsf{R}(y) = y]$, where the distribution of $y \sim \mathbf{p}$ is $\Pr[y = i] = p_i$ for all $i \in [K]$.

The main result of this section is that, among all $\varepsilon$-DP algorithms, `RRWithPrior` maximizes $\mathrm{Obj}_{\mathbf{p}}(\mathsf{R})$, as stated more formally next.

**Theorem 3.** *Let* $\mathbf{p}$ *be any probability distribution on* $[K]$ *and* R *be any* $\varepsilon$-*DP algorithm that randomizes the input label given the prior* $\mathbf{p}$ *. We have that*

$$\mathrm{Obj}_{\mathbf{p}}(\texttt{RRWithPrior}) \geq \mathrm{Obj}_{\mathbf{p}}(\mathsf{R}).$$

Before we proceed to the proof, we remark that our proof employs a linear program (LP) to characterize the optimal mechanisms; a generic form of such LPs has been used before in [48, 41]. However, these works focus on different problems (linear queries) and their results do not apply here.

*Proof of Theorem 3.* Consider any $\varepsilon$-DP algorithm R, and let $q_{\tilde{y}|y}$ denote $\Pr[\mathsf{R}(y) = \tilde{y}]$. Observe that $\mathrm{Obj}_{\mathbf{p}}(\mathsf{R}) = \sum_{y \in [k]} p_y \cdot q_{y|y}$.

Since $q_{\cdot|y}$ is a probability distribution, we must have that

$$\sum_{\tilde{y} \in [K]} q_{\tilde{y}|y} = 1, \forall y \in [K], \text{ and } q_{\tilde{y}|y} \geq 0, \forall \tilde{y}, y \in [K].$$

Finally, the $\varepsilon$-DP guarantee of R implies that

$$q_{\tilde{y}|y} \leq e^{\varepsilon} \cdot q_{\tilde{y}|y'} \qquad\qquad \forall \tilde{y}, y, y' \in [K].$$

Combining the above, $\mathrm{Obj}_{\mathbf{p}}(\mathsf{R})$ is upper-bounded by the optimum of the following linear program (LP), which we refer to as LP1:

$$\begin{aligned}
\max \quad & \sum_{y \in [k]} p_y \cdot q_{y|y} \\
\text{s.t.} \quad & q_{\tilde{y}|y} \leq e^{\varepsilon} \cdot q_{\tilde{y}|y'} & \forall \tilde{y}, y, y' \in [K], & \quad (2) \\
& \sum_{\tilde{y} \in [K]} q_{\tilde{y}|y} = 1 & \forall y \in [K], & \quad (3) \\
& q_{\tilde{y}|y} \geq 0 & \forall \tilde{y}, y \in [K]. &
\end{aligned}$$

Notice that constraints (2) and (3) together imply that:

$$q_{y|y} + e^{-\varepsilon} \cdot \sum_{\tilde{y} \in [K] \setminus \{y\}} q_{\tilde{y}|\tilde{y}} \leq 1 \qquad\qquad \forall y \in [K].$$

In other words, the optimum of LP1 is at most the optimum of the following LP that we call LP2:

$$\begin{aligned}
\max \quad & \sum_{y \in [k]} p_y \cdot q_{y|y} \\
\text{s.t.} \quad & q_{y|y} + e^{-\varepsilon} \cdot \sum_{\tilde{y} \in [K] \setminus \{y\}} q_{\tilde{y}|\tilde{y}} \leq 1 & \forall y \in [K], & \quad (4) \\
& q_{y|y} \geq 0 & \forall y \in [K]. & \quad (5)
\end{aligned}$$

An optimal solution to LP2 must be a vertex (aka extreme point) of the polytope defined by (4) and (5). Recall that an extreme point of a $K$-dimensional polytope must satisfy $K$ independent constraints with equality. In our case, this means that one of the following occurs:

- Inequality (5) is satisfied with equality for all $y \in [K]$ resulting in the all-zero solution (whose objective is zero), or,
- For some non-empty subset $Y \subseteq [K]$, inequality (4) is satisfied with equality for all $y \in Y$, and inequality (5) is satisfied with equality for all $y \in [K] \setminus Y$. This results in

$$q_{y|y} = \begin{cases} \frac{e^{\varepsilon}}{e^{\varepsilon} + |Y| - 1} & \text{if } y \in Y, \\ 0 & \text{if } y \notin Y. \end{cases}$$

This yields an objective value of $\frac{e^{\varepsilon}}{e^{\varepsilon} + |Y| - 1} \cdot \sum_{y \in Y} p_y$.

In conclusion, we have that

$$\begin{aligned}
\mathrm{Obj}_{\mathbf{p}}(\mathsf{R}) &\leq \max_{\emptyset \neq Y \subseteq [K]} \frac{e^{\varepsilon}}{e^{\varepsilon} + |Y| - 1} \cdot \sum_{y \in Y} p_y \\
&= \max_{k \in [K]} \frac{e^{\varepsilon}}{e^{\varepsilon} + k - 1} \cdot \max_{Y \subseteq [K], |Y| = k} \sum_{y \in Y} p_y \\
&= \max_{k \in [K]} \frac{e^{\varepsilon}}{e^{\varepsilon} + k - 1} \cdot \sum_{y \in Y_k} p_y = \max_{k \in [K]} w_k,
\end{aligned}$$

where the last two equalities follow from our definitions of $Y_k$ and $w_k$. Notice that $\mathrm{Obj}_{\mathbf{p}}(\mathsf{RRWithPrior}) = \max_{k \in [K]} w_k$. Thus, we get that $\mathrm{Obj}_{\mathbf{p}}(\mathsf{RRWithPrior}) \geq \mathrm{Obj}_{\mathbf{p}}(\mathsf{R})$ as desired. $\qquad\square$

# 4 Application of `RRWithPrior`: Multi-Stage Training

Our `RRWithPrior` algorithm requires publicly available priors, which could usually be obtained from domain specific knowledge. In this section, we describe a training framework that bootstraps from a uniform prior, and progressively learns refined priors via multi-stage training. This general framework can be applied to arbitrary domains even when no public prior distributions are available.

Specifically, we assume that we have a training algorithm `A` that outputs a probabilistic classifier which, on a given unlabeled sample $\mathbf{x}$, can assign a probability $p_y$ to each class $y \in [K]$. We partition our dataset into subsets $S^{(1)}, \ldots, S^{(T)}$, and we start with a trivial model $M^{(0)}$ that outputs equal probabilities for all classes. At each stage $t \in [T]$, we use the most recent model $M^{(t-1)}$ to assign the probabilities $(p_1, \ldots, p_K)$ for each sample $\mathbf{x}_i$ from $S^{(t)}$. Applying `RRWithPrior` with this prior on the true label $y_i$, we get a randomized label $\tilde{y}_i$ for $\mathbf{x}_i$. We then use all the samples with randomized labels obtained so far to train the model $M^{(t)}$.

The full description of our LP-MST (Label Privacy Multi-Stage Training) method is presented in Algorithm 3. We remark here that the partition $S^{(1)}, \ldots, S^{(T)}$ can be arbitrarily chosen, as long as it does not depend on the labels $y_1, \ldots, y_n$. We also stress that the training algorithm `A` need *not* be private. We use `LP-1ST` to denote our algorithm with one stage, `LP-2ST` to denote our algorithm with two stages, and so on. We also note that `LP-1ST` is equivalent to using vanilla RR. The $t$th stage of a multi-stage algorithm is denoted *stage-t*.

---

**Algorithm 3 `Multi-Stage Training (LP-MST)`**

**Input:** Dataset $S = \{(\mathbf{x}_1, y_1), \ldots, (\mathbf{x}_n, y_n)\}$
**Parameters:** Number $T$ of stages, training algorithm `A`

1. Partition $S$ into $S^{(1)}, \ldots, S^{(T)}$
2. Let $M^{(0)}$ be the trivial model that always assigns equal probability to each class.
3. For $t = 1$ to $T$:
    (a) Let $\tilde{S}^{(t)} = \emptyset$.
    (b) For each $(\mathbf{x}_i, y_i) \in S^{(t)}$:
        i. Let $\mathbf{p} = (p_1, \ldots, p_K)$ be the probabilities predicted by $M^{(t)}$ on $\mathbf{x}_i$.
        ii. Let $\tilde{y}_i = \texttt{RRWithPrior}_{\mathbf{p}}(y_i)$.
        iii. Add $(\mathbf{x}_i, \tilde{y}_i)$ to $\tilde{S}^{(t)}$.
    (c) Let $M^{(t)}$ be the model resulting from training on $\tilde{S}^{(1)} \cup \cdots \cup \tilde{S}^{(t)}$ using `A`.
4. Output $M^{(T)}$.

---

The privacy guarantee of LP-MST is given by the following:

**Observation 4.** *For any $\varepsilon > 0$, if `RRWithPrior` is $\varepsilon$-DP, then `LP-MST` is $\varepsilon$-LabelDP.*

*Proof.* We will in fact prove a stronger statement that the algorithm is $\varepsilon$-DP even when we output all the $T$ models $M^{(1)}, \ldots, M^{(T)}$ together with all the randomized labels $\tilde{y}_1, \ldots, \tilde{y}_n$. For any possible output models $m^{(1)}, \ldots, m^{(T)}$ and output labels $z_1, \ldots, z_n$, we have

$$\Pr[M^{(1)} = m^{(1)}, \ldots, M^{(T)} = m^{(T)}, \tilde{y}_1 = z_1, \ldots, \tilde{y}_n = z_n]$$
$$= \prod_{t=1}^{T} \left( \Pr\left[ M^{(t)} = m^{(t)} \middle| \bigwedge_{i \in S^{(1)} \cup \cdots \cup S^{(t)}} \tilde{y}_i = z_i \right] \cdot \prod_{i \in S^{(t)}} \Pr\left[ \tilde{y}_i = z_i \middle| M^{(t-1)} = m^{(t-1)} \right] \right).$$

Consider any two datasets $\mathbf{D}, \mathbf{D}'$ that differ on a single user's label; suppose this user is $j$ and that the user belongs to partition $\ell \in [T]$. Then, the above expression for $\mathbf{D}$ and that for $\mathbf{D}'$ are the same in all but one term: $\Pr\left[ \tilde{y}_j = z_j \middle| M^{(\ell-1)} = m^{(\ell-1)} \right]$, which is the probability that $\texttt{RRWithPrior}_{m^{(\ell-1)}(x_i)}$ outputs $z_i$. Since `RRWithPrior` is $\varepsilon$-DP, we can conclude that the ratio between the two probabilities is at most $e^\varepsilon$ as desired. $\square$

We stress that this observation holds because each sensitive label $y_i$ is only used *once* in Line 3(b)ii of Algorithm 3, as the dataset $S$ is partitioned at the beginning of the algorithm. As a result, since each

Table 1: Test accuracy (%) on CIFAR-10. The baseline performances taken from previously published results correspond to $(\varepsilon, \delta)$-DP with $\delta = 10^{-5}$. The star$^\star$ indicates the use of CIFAR-100 pre-trained representations.

| Algorithm | $\varepsilon = 1$ | $\varepsilon = 2$ | $\varepsilon = 3$ | $\varepsilon = 4$ | $\varepsilon = 6$ | $\varepsilon = 8$ | $\varepsilon = \infty$ |
|---|---|---|---|---|---|---|---|
| `DP-SGD` w/ pre-train$^\star$ [2] | | 67 | | 70 | | 73 | 80 |
| `DP-SGD` [77] | | | | | | $61.6_{(\varepsilon=7.53)}$ | 76.6 |
| Tempered Sigmoid [77] | | | | | | $66.2_{(\varepsilon=7.53)}$ | |
| Yu et al. [98] | | | | | $44.3_{(\varepsilon=6.78)}$ | | |
| Nasr et al. [71] | | | 55 | | | | |
| Chen and Lee [20] | | | | | | 53 | |
| ScatterNet+CNN [93] | | | 69.3 | | | | |
| `LP-1ST` | 59.96 | 82.38 | 89.89 | 92.58 | 93.58 | 94.70 | 94.96 |
| `LP-2ST` | 63.67 | 86.05 | 92.19 | 93.37 | 94.26 | 94.52 | - |
| `LP-1ST` w/ pre-train$^\star$ | 67.64 | 83.99 | 90.24 | 92.83 | 94.02 | 94.96 | 95.25 |
| `LP-2ST` w/ pre-train$^\star$ | 70.16 | 87.22 | 92.12 | 93.53 | 94.41 | 94.59 | - |

Table 2: Experiments on CIFAR-100. The non-private baseline ($\varepsilon = \infty$) is 76.38% test accuracy.

| Algorithm | $\varepsilon = 3$ | $\varepsilon = 4$ | $\varepsilon = 5$ | $\varepsilon = 6$ | $\varepsilon = 8$ |
|---|---|---|---|---|---|
| `LP-1ST` | 20.96 | 46.28 | 61.38 | 68.34 | 73.59 |
| `LP-2ST` | 28.74 | 50.15 | 63.51 | 70.58 | 74.14 |

stage is $\varepsilon$-`LabelDP`, the entire algorithm is also $\varepsilon$-`LabelDP`. This is known as (an adaptive version of a) *parallel composition* [68].

Finally, we point out that the running time of our `RRWithPrior` algorithm is quasi-linear in $K$ (the time needed to sort the prior). This is essentially optimal within multistage training, since $O(K)$ time will be required to write down the prior after each stage. Moreover, for reasonable values of $K$, the running time will be dominated by back-propagation for gradient estimation. Moreover, the focus of the current work is on small to modest label spaces (i.e., values of $K$).

## 5 Empirical Evaluation

We evaluate `RRWithPrior` on standard benchmark datasets that have been widely used in previous works on private machine learning. Specifically, in the first part, we study our general multi-stage training algorithm that boostraps from a uniform prior. We evaluate it on image classification and collaborative filtering tasks. In the second part, we focus on image classification only and use domain-specific techniques to obtain priors for `RRWithPrior`. We use modern neural network architectures (e.g., ResNets [51]) and the mixup [101] regularization for learning with noisy labels. Please see the Supplementary Material for full details on the datasets and the experimental setup.

### 5.1 Evaluation with Multi-Stage Training

CIFAR-10 [60] is a 10-class image classification benchmark dataset. We evaluate our algorithm and compare it to previously reported DP baselines in Table 1. Due to scalability issues, previous DP algorithms could only use simplified architectures with non-private accuracy significantly below the state-of-the-art. Moreover, even when compared to those weaker non-private baselines, a large performance drop is observed in the private models. In contrast, we use ResNet18 with 95% non-private accuracy. Overall, our algorithms improve the previous state-of-the-art by a margin of 20% across all $\varepsilon$'s. Abadi et al. [2] treated CIFAR-100 as public data and use it to pre-train a representation to boost the performance of `DP-SGD`. We also observe performance improvements with CIFAR-100 pre-training (Table 1, bottom 2 rows). But even without pre-training, our results are significantly better than `DP-SGD` even *with* pre-training.

Table 3: Experiments on MovieLens-1M. The numbers show the test RMSE.

| Algorithm | $\varepsilon = 1$ | $\varepsilon = 2$ | $\varepsilon = 3$ | $\varepsilon = 4$ | $\varepsilon = 8$ | $\varepsilon = \infty$ |
|---|---|---|---|---|---|---|
| LP-1ST | 1.122 | 0.981 | 0.902 | 0.877 | 0.867 | 0.868 |
| LP-2ST | 1.034 | 0.928 | 0.891 | 0.874 | 0.865 | |
| Gaussian DP [15] | | | | | | 0.915 ($\varepsilon \geq 10$) |

In Table 2 we also show results on CIFAR-100, which is a more challenging variant with $10\times$ more classes. To the best of our knowledge, these are the first non-trivial reported results on CIFAR-100 for DP learning. For $\varepsilon = 8$, our algorithm is only 2% below the non-private baseline.

In addition, we also evaluate on MovieLens-1M [49], which contains 1 million anonymous ratings of approximately $3,900$ movies, made by 6,040 MovieLens users. Following [15], we randomly split the data into $80\%$ train and $20\%$ test, and show the test Root Mean Square Error (RMSE) in Table 3.

Results on MNIST [61], Fashion MNIST [97], and KMNIST [25], and comparison to more baselines can be found in the Supplementary Material. In all the datasets we evaluated, our algorithms not only significantly outperform the previous methods, but also greatly shrink the performance gap between private and non-private models. The latter is critical for applications of deep learning systems in real-world tasks with privacy concerns.

**Beyond Two Stages.** In Figure 1(a), we report results on LP-MST with $T > 2$. For the cases we tried, we consistently observe 1–2% improvements on test accuracy when going from LP-2ST to LP-3ST. In our preliminary experiments, going beyond $T > 4$ stages leads to diminishing returns on some datasets.

## 5.2 Evaluation with Domain-Specific Priors

The multi-stage training framework evaluated in the previous section is a general domain-agnostic algorithm that bootstraps itself from uniform priors. In some cases, domain-specific priors can be obtained to further improve the learning performance. In this section, we focus on image classification applications, where new advances in self-supervised learning (SSL) [22, 23, 44, 53, 16] show that high-quality image representations could be learned on large image datasets without using the class labels. In the setting of LabelDP, the unlabeled images are considered public data, so we design an algorithm to use SSL to obtain priors, which is then fed to RRWithPrior for discriminative learning.

Specifically, we partition the training examples into groups by clustering using their representations extracted from SSL models. We then query a histogram of labels for each group via discrete Laplace mechanism (aka Geometric Mechanism) [41]. If the groups are largely homogeneous, consisting of mostly examples from the same class, then we can make the histogram queries with minimum privacy budget. The queried histograms are used as label priors for all the points in the group. Figure 1(b) shows the results on two different SSL representations: BYOL [44], trained on unlabeled CIFAR-10 images and DINO [16], trained on ImageNet [27] images. Comparing to the baseline, the SSL-based priors significantly improves the model performance with small privacy budgets. Note that since the SSL priors are not *true* priors, with large privacy budget ($\varepsilon = 8$), it actually underperforms the uniform prior. But in most real world applications, small $\varepsilon$'s are generally more useful.

## 6 Theoretical Analysis

Previous works have shown that LabelDP can be provably easier than DP in certain settings; specifically, in the PAC learning setting, Beimel et al. [13] proved that finite VC dimension implies learnability by LabelDP algorithms, whereas it is known that this is not sufficient for DP algorithms [7].

We extend the theoretical understanding of this phenomenon to the stochastic convex optimization (SCO) setting. Specifically, we show that, by applying RR on the labels and running SGD on top of the resulting noisy dataset with an appropriate debiasing of the noise, one can arrive at the following *dimension-independent* excess population loss.

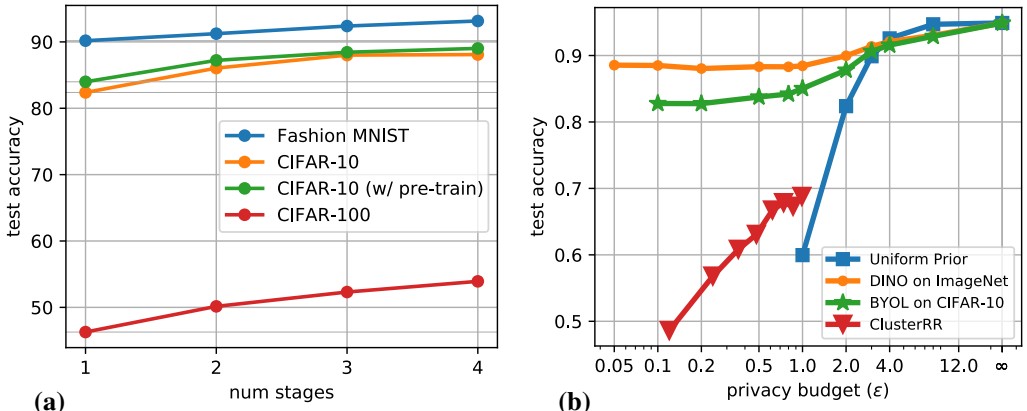

Figure 1: (a) Test accuracy (%) on various datasets with `LP-MST` for $T > 2$. The curve "CIFAR-10 w/ pre-train" is using CIFAR-100 as public data to pre-train the model. (b) `RRWithPrior` with priors obtained from histogram query based on clustering in various SSL representations. We also plot recent results (ClusterRR) from Esfandiari et al. [37], which is a clustering based `LabelDP` algorithm.

**Theorem 5** (Informal). *For any $\varepsilon \in (0, 1)$, there is an $\varepsilon$-`LabelDP` algorithm for stochastic convex optimization with excess population loss $\tilde{O}\left(DL \cdot \frac{K}{\varepsilon\sqrt{n}}\right)$ where $D$ denotes the diameter of the parameter space and $L$ denotes the Lipschitz constant of the loss function.*

The above excess population loss can be compared to that of Bassily et al. [12], who gave an $(\varepsilon, \delta)$-DP algorithm with excess population loss $O_{D,L}\left(\frac{1}{\sqrt{n}} + \frac{\sqrt{p}}{\varepsilon n}\right)$, where $p$ denote the dimension of the parameter space; this bound is also known to be tight in the standard DP setting. The main advantage of our guarantee in Theorem 5 is that it is independent of the dimension $p$. Furthermore, we show that our bound is tight up to polylogarithmic factors and the dependency on the number of classes $K$.

The above result provides theoretical evidence that running `RR` on the labels and then training on this noisy dataset can be effective. We can further extend this to the setting where, instead of running `RR`, we run `RRTop-`$k$ before running the aforementioned (debiased) SGD, although—perhaps as expected—our bound on the population loss now depends on the quality of the priors.

**Corollary 6** (Informal). *Suppose that we are given a prior $\mathbf{p}_x$ for every $x$ and let $Y_k^x$ denote the set of top-$k$ labels with respect to $\mathbf{p}_x$. Then, for any $\varepsilon \in (0, 1)$, there is an $\varepsilon$-`LabelDP` algorithm for stochastic convex optimization with excess population loss $\tilde{O}\left(DL \cdot \left(\frac{k}{\varepsilon\sqrt{n}} + \Pr_{(x,y)\sim\mathcal{D}}[y \notin Y_k^x]\right)\right)$ where $D, L$ are as defined in Theorem 5 and $\mathcal{D}$ is the data distribution.*

When our top-$k$ set is perfect (i.e., $y$ always belongs to $Y_k^x$), the bound reduces to that of Theorem 5, but with the smaller $k$ instead of $K$. Moreover, the second term is, in some sense, a penalty we pay in the excess population loss for the inaccuracy of the top-$k$ prior. We defer the formal treatment and the proofs to the Supplementary Material, in which we also present additional generalization results for non-convex settings.

Note that Corollary 6 is not in the exact setup we run in experiments, where we dynamically calculate an optimal $k$ for each $x$ given generic priors (via `RRWithPrior`), and for which the utility is much more complicated to analyze mathematically. Nonetheless, the above corollary corroborates the intuition that a good prior helps with training.

## 7   Conclusions and Future Directions

In this work, we introduced a novel algorithm `RRWithPrior` (which can be used to improve on the traditional RR mechanism), and applied it to `LabelDP` problems. We showed that prior information can be incorporated to the randomized label querying framework while maintaining privacy constraints. We demonstrated two frameworks to apply `RRWithPrior`: (i) a general multi-stage training algorithm `LP-MST` that bootstraps from uniform priors and (ii) an algorithm that build priors from clustering with

SSL-based representations. The former is general purpose and can be applied to tasks even when no domain-specific priors are available, while the latter uses a domain-specific algorithm to extract priors and performs well even with very small privacy budget. As summarized by the figure on the right, in both cases, by focusing on LabelDP, our RRWithPrior significantly improved the model performance of previous state-of-the-art DP models that aimed to protect both the inputs and outputs. We note that, following up on our work, additional results on deep learning with LabelDP were obtained [66, 100]. The narrowed performance gap between private and non-private models is vital for adding DP to real world deep learning models. We nevertheless stress that our algorithms only protect the labels but not the input points, which might not constitute a sufficient privacy protection in all settings.

Our work opens up several interesting questions. Firstly, note that our multi-stage training procedure uses very different ingredients than those of Abadi et al. [2] (which employ DP-SGD, privacy amplification by subsampling, and Renyi accounting); can these tools be used to further improve LabelDP? Secondly, while our procedure can be implemented in the most stringent *local* DP setting[5] [58], can it be improved in the weaker central (aka trusted curator) DP model, assuming the curator knows the prior? Thirdly, while our algorithm achieves pure DP (i.e., $\delta = 0$), is higher accuracy possible for approximate DP (i.e., $\delta > 0$)?

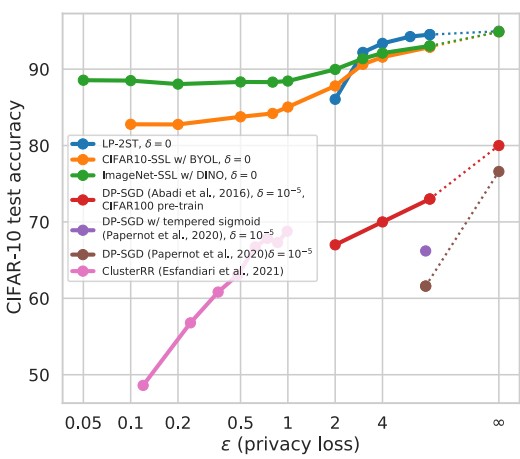

# Acknowledgements

The authors would like to thank Sami Torbey for very helpful feedback on an early version of this work. At MIT, Noah Golowich was supported by a Fannie and John Hertz Foundation Fellowship and an NSF Graduate Fellowship.

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
