# Supplementary Material for "Deep Learning with Label Differential Privacy"

## A  Missing Proofs

### A.1  Proof of Lemma 1

*Proof of Lemma 1.* Consider any inputs $y, y' \in [K]$ and any possible output $\tilde{y} \in Y_k$. $\Pr[\text{RRTop-}k(y) = \tilde{y}]$ is maximized when $y = \tilde{y}$, whereas $\Pr[\text{RRTop-}k(y') = \tilde{y}]$ is minimized when $y' \in Y_k \setminus \{\tilde{y}\}$. This implies that

$$\frac{\Pr[\text{RRTop-}k(y) = \tilde{y}]}{\Pr[\text{RRTop-}k(y') = \tilde{y}]} \leq \frac{\frac{e^\varepsilon}{e^\varepsilon + k - 1}}{\frac{1}{e^\varepsilon + k - 1}} = e^\varepsilon.$$

Thus, RRTop-$k$ is $\varepsilon$-DP as desired. $\qquad\square$

## B  Details of the Experimental Setup

**Datasets.**  We evaluate our algorithms on the following image classification datasets:

- MNIST [61], 10 class classification of hand written digits, based on inputs of $28 \times 28$ gray scale images. The training set contains 60,000 examples and the test set contains 10,000.
- Fashion MNIST [97], 10 class classification of Zalando's article images. The dataset size and input format are the same as MNIST.
- KMNIST [25], 10 class classification of Hiragana characters. The dataset size and the input format are the same as MNIST.
- CIFAR-10/CIFAR-100 [60] are 10 class and 100 class image classification datasets, respectively. Both datasets contains $32 \times 32$ color images, and both have a training set of size 50,000 and a test set of size 10,000.
- MovieLens [49] contains a set of movie ratings from the MovieLens users. It was collected and maintained by a research group (GroupLens) at the University of Minnesota. There are 5 versions: "25m", "latest-small", "100k", "1m", "20m". Following Bu et al. [15], we use the "1m" version, which the largest MovieLens dataset that contains demographic data. Specifically, it contains 1,000,209 anonymous ratings of approximately 3,900 movies made by 6,040 MovieLens users, with some meta data such as gender and zip code.

**Architectures.**  On CIFAR-10/CIFAR-100, we use ResNet [51], which is a Residual Network architecture widely used in the computer vision community. In particular, we use ResNet18 V2 [52]. Note the standard ResNet18 is originally designed for ImageNet scale (image size $224 \times 224$). When adapting to CIFAR (image size $32 \times 32$), we replace the initial block with $7 \times 7$ convolution and $3 \times 3$ max pooling with a single $3 \times 3$ convolution (with stride 1) layer. The upper layers are kept the same as the standard ImageNet ResNet18. On MNIST, Fashion MNIST, and KMNIST, we use a simplified Inception [91] model suitable for small image sizes, and defined as follows:

$$
\begin{aligned}
\text{Inception ::} \quad & \text{Conv}(3{\times}3, 96) \to \text{S1} \to \text{S2} \to \text{S3} \to \text{GlobalMaxPool} \to \text{Linear}. \\
\text{S1 ::} \quad & \text{Block}(32, 32) \to \text{Block}(32, 48) \to \text{Conv}(3{\times}3, 160, \text{Stride=2}). \\
\text{S2 ::} \quad & \text{Block}(112, 48) \to \text{Block}(96, 64) \to \text{Block}(80, 80) \to \text{Block }(48, 96) \to \\
& \text{Conv}(3{\times}3, 240, \text{Stride=2}). \\
\text{S3 ::} \quad & \text{Block}(176, 160) \to \text{Block}(176, 160). \\
\text{Block}(C_1, C_2) \text{ ::} \quad & \text{Concat}(\text{Conv}(1{\times}1, C_1), \text{Conv}(3{\times}3, C_2)). \\
\text{Conv ::} \quad & \text{Convolution} \to \text{BatchNormalization} \to \text{ReLU}.
\end{aligned}
$$

For the MovieLens experiment, we adopt a two branch neural networks from the neural collaborative filtering algorithm [54]. We simply treat the ratings as categorical labels and apply our algorithm for

multi-class classification. During evaluation, we output the average rating according to the softmax probabilities output by the trained model.

**Training Procedures.** On MNIST, Fashion MNIST, and KMNIST, we train the models with mini-batch SGD with batch size 265 and momentum 0.9. We run the training for 40 epochs (for multi-stage training, each stage will run 40 epochs separately), and schedule the learning rate to linearly grow from 0 to 0.02 in the first 15% training iterations, and then linearly decay to 0 in the remaining iterations.

On CIFAR-10, we use batch size 512 and momentum 0.9, and train for 200 epochs. The learning rate is scheduled according to the widely used *piecewise constant with linear rampup* scheme. Specifically, it grows from 0 to 0.4 in the first 15% training iterations, then it remains piecewise constant with a decay factor of 10 at the 30%, 60%, and 90% training iterations, respectively. The CIFAR-100 setup is similar to CIFAR-10 except that we use a batch size 256 and a peak learning rate 0.2. MovieLens experiments are trained similarly, but with batch size 128.

On all datasets, we optimize the cross entropy loss with an $\ell_2$ regularization (coefficient $10^{-4}$). All the networks are randomly initialized at the beginning of the training. For the experiment on CIFAR-10 where we explicitly study the effect of pre-training to compare with previous methods that use the same technique, we train a (non-private) ResNet18 on the full CIFAR-100 training set and initialize the CIFAR-10 model with the pre-trained weights. The classifier is still randomly initialized because there is no clear correspondence between the 100 classes of CIFAR-100 and the 10 classes of CIFAR-10. The remaining configuration remains the same as in the experiments without pre-training. In particular, we did *not* freeze the pre-trained weights.

We apply standard data augmentations, including random crop, random left-right flip, and random cutout [28], to all the datasets during training. We implement our algorithms in TensorFlow [1], and train all the models on NVidia Tesla P100 GPUs.

**Learning with Noisy Labels.** Standard training procedures tend to overfit to the label noise and generalize poorly on the test set when some of the training labels are randomly flipped. We apply *mixup* [101] regularization, which generates random convex combinations of both the inputs and the (one-hot encoded) labels during training. It is shown that mixup is resistant to random label noise. Note that our framework is generic and in principle any robust training technique could be used. We have chosen mixup for its simplicity, but there has been a rich body of recent work on deep learning methods with label noise, see, e.g., [55, 46, 99, 21, 104, 74, 69, 64, 105, 56, 50, 47, 65, 86, 79, 87] and the references therein. Potentially with more advanced robust training, even higher performance could be achieved.

**Multi-Stage Training.** There are a few implementation enhancements that we find useful for multi-stage training. For concreteness, we discuss them for LP-2ST. First, we find it helps to initialize the stage-2 training with the models trained in stage-1. This is permitted as the stage-1 model is trained on labels that are queried privately. Moreover, we can reuse those labels queried in stage-1 and train stage-2 on a combined dataset. Although the subset of data from stage-1 is noisier, we find that it generally helps to have more data, especially when we reduce the noise of stage-1 data by using the learned prior model. Specifically, for each sample $(x, \tilde{y})$ in the stage-1 data, where $\tilde{y}$ is the private label queried in stage-1, we make a prediction on $x$ using the model trained in stage-1; if $\tilde{y}$ is not in the top $k$ predicted classes, we will exclude it from the stage-2 training. Here $k$ is simply set to the average $k$ obtained when running RRWithPrior to query labels on the data held out for stage-2. Similar ideas apply to training with more stages. For example, in LP-3ST, stage-3 training could use the model trained in stage-2 as initialization, and use it to filter the queried labels in stage-1 and stage-2 that are outside the top $k$ prediction, and then train on the combined data of all 3 stages.

**Priors from Self-supervised Learning.** Recent advances in self-supervised learning (SSL) [22, 23, 44, 53, 16] show that representations learned from a large collection of unlabeled but diverse images could capture useful semantic information and can be finetuned with labels to achieve classification performance on par with the state-of-the-art fully supervised learned models. We apply SSL algorithms to extract priors for image classification problems, with the procedure described in Algorithm 4.

Specifically, we choose two recent SSL algorithms: BYOL [44] and DINO [16]. For BYOL, we train the SSL model using the (unlabeled) CIFAR-10 images only, as a demonstration without using

**Algorithm 4** SSL Priors.

**Input:** Training set $D = \{(x_i, y_i)\}_{i=1}^n$, cluster count $C$, privacy budget for priors $\varepsilon_p$, trained SSL model $f_{\text{SSL}}$.

1. Initialize $P \leftarrow 1/K$ ones$(n, K)$ as the uniform priors.
2. Extract SSL features $F = \{f_{\text{SSL}}(x_i) : (x_i, y_i) \in D\}$.
3. Run $k$-means algorithms to partition $F$ into $C$ groups.
4. For each $c = 1$ to $C$:
    (a) Compute histogram of classes $H_c \in \mathbb{N}_{\geq 0}^K$ according to the labels of examples in the $c$-th group.
    (b) Get a private histogram query $\tilde{H}_c \leftarrow H_c +$ `scipy.stats.dlaplace.rvs`$(\varepsilon_p/2, K)$, via the discrete Laplace mechanism.
    (c) Get a prior via normalization: $p_c = \max(\tilde{H}_c, 0) / \sum_{k=1}^K \max(\tilde{H}_c[k], 0)$.
    (d) For each example $i$ in group $c$, assign $P[i, :] \leftarrow p_c$.
5. Output $P$.

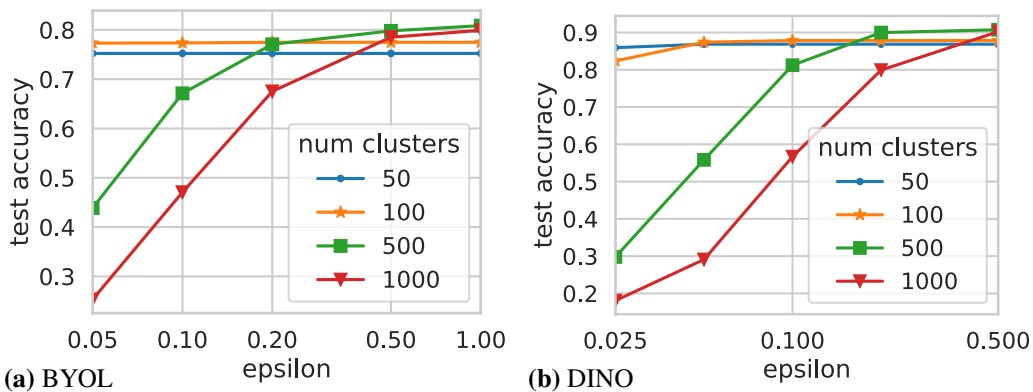

**(a)** BYOL          **(b)** DINO

Figure 2: Accuracy evaluated on CIFAR-10 test set, of private histogram querying with kmeans clustering on self-supervised learning based features learned by (a) BYOL [44] on CIFAR-10 and (b) DINO [16] on ImageNet.

*external* data. For DINO, we use the models pre-trained on (unlabeled) ImageNet [27] images. Since ImageNet is a much larger and more diverse dataset than CIFAR-10, the SSL representations are also more capable of capturing the semantic information. Note the ImageNet images are of higher resolution and resized to $224 \times 224$ during training. To extract features for $32 \times 32$ CIFAR-10 images, we simply upscale the images to $224 \times 224$ before feeding into the trained neural network.

We choose relatively large cluster sizes so that the private histogram query is more robust to the added discrete Laplace noise. In particular, we found $C = 100$ clusters for BYOL representations and $C = 50$ clusters for DINO representations achieve a good balance of robustness and accuracy. Since $\varepsilon_p$ will be subtracted from the privacy budget for `RRWithPrior`, we simply choose the smallest $\varepsilon_p$ without causing too much deterioration of the priors. In our experiments, we set $\varepsilon_p = 0.05$ for BYOL and $\varepsilon_p = 0.025$ for DINO. Note the model accuracy could potentially be further boosted by choosing $C$ and $\varepsilon_p$ adaptively according to the overall privacy budget. In the following, we provide a simple study to show how the interplay between $\varepsilon_p$ and $C$ affects the accuracy of the histogram queries.

To compute an accuracy measure on the test set, we extract features using a SSL learned models on both training and test set. A $k$-means clustering algorithm is run on the joint set of training and test features. For each cluster, we apply the discrete Laplace mechanism to make a private histogram of class distributions from *only the training examples* in that cluster. The class with the maximum votes are then used as predicted labels for all the *test examples* in the cluster, and compared with the true test labels to calculate the accuracy. Figure 2 shows the accuracy with the two different SSL features under different privacy budgets ($\varepsilon$) for making the histogram queries. As expected, the accuracy is higher with smaller clusters, but at the same time sensitive to noise introduced by the Geometric Mechanism when the privacy budget is small.

Table 4: Test accuracy (%) on MNIST and Fashion MNIST. The baseline performances taken from previously published results correspond to $(\varepsilon, \delta)$-DP with $\delta = 10^{-5}$.

| | Algorithm | $\varepsilon = 1$ | $\varepsilon = 2$ | $\varepsilon = 3$ | $\varepsilon = 4$ | $\varepsilon = 8$ | $\varepsilon = \infty$ |
|---|---|---|---|---|---|---|---|
| **MNIST** | DP-SGD [2] | | 95 | | | 97 | 98.3 |
| | PATE-G [75] | | 98($\varepsilon$=2.04) | | | 98.1( $\varepsilon$=8.03) | 99.2 |
| | Confident-GNMax [76] | | 98.5($\varepsilon$=1.97) | | | | 99.2 |
| | Tempered Sigmoid [77] | | 98.1($\varepsilon$=2.93) | | | | |
| | Bu et al. [15] | | 96.6($\varepsilon$=2.32) | | | 97.0($\varepsilon$ =5.07) | |
| | Chen and Lee [20] | | 90.0($\varepsilon$ =2.5) | | | | |
| | Nasr et al. [71] | | | 96.1($\varepsilon$ =3.2) | | | |
| | Yu et al. [98] | | | | 93.2($\varepsilon$ =6.78) | | |
| | Feldman and Zrnic [39] | 96.56($\varepsilon$=1.2) | | 97.71 | | | |
| | LP-1ST | 95.34 | 98.16 | 98.81 | 99.08 | | 99.33 |
| | LP-2ST | 95.82 | 98.78 | 99.14 | 99.24 | | |
| **Fashion MNIST** | DP-SGD [77] | | 81.9($\varepsilon$=2.7) | | | | 89.4 |
| | Tempered Sigmoid [77] | | 86.1($\varepsilon$=2.7) | | | | |
| | Chen and Lee [20] | | 82.3 | | | | |
| | LP-1ST | 80.78 | 90.18 | 92.52 | 93.50 | | 94.28 |
| | LP-2ST | 83.26 | 91.24 | 93.18 | 94.10 | | |

Table 5: Test accuracy (%) on KMNIST [25].

| Algorithm | $\varepsilon$=1 | $\varepsilon$=2 | $\varepsilon$=3 | $\varepsilon$=4 | $\varepsilon$=$\infty$ |
|---|---|---|---|---|---|
| LP-1ST | 76.56 | 92.04 | 95.86 | 96.86 | 98.33 |
| LP-2ST | 81.26 | 93.72 | 97.19 | 97.83 | - |

## C Extra Results on Multi-Stage Training

In addition to the results presented in the main text, we include extra results of multi-stage training on MNIST [61], Fashion MNIST [97], and KMNIST [25]. Both MNIST and Fashion MNIST have been previously used to benchmark DP deep learning algorithms. We compare our algorithms with previously reported numbers in Table 4. Our algorithms outperform previous methods across all $\varepsilon$'s on both datasets. The gap is more pronounced on Fashion MNIST, which is slightly harder than MNIST. Furthermore, LP-2ST consistently improves over LP-1ST. Table 5 shows the model performances on KMNIST under different privacy losses. The results are qualitatively similar to the ones for MNIST and Fashion MNIST.

## D Learning Dynamics of Multi-stage Training

Fig. 3 visualizes the learning curves of LP-1ST and LP-2ST on CIFAR-10 with $\varepsilon = 2$. Stage-1 of LP-2ST (using 65% training data) clearly underperforms LP-1ST with the full training set. But it is good enough to provide useful prior for stage-2. The RRWithPrior algorithm responds with an average $k = 1.86$ over the remaining 35% of the training set. As the dotted line shows, the top-2 accuracy of the model trained in stage-1 reaches 90% at the end of training, indicating that the true label on the test set is within the top-2 prediction with high probability. In stage-2, we continue with the model trained in stage-1, and train on the combined data of the two stages. This is possible because the labels queried in stage-1 are already private. As a result, LP-2ST achieves higher performance than LP-1ST.

## E Analysis of Robustness to Hyperparameters

Following previous work, [e.g., 77], we report the benchmark performance after hyperparameter tuning. In practice, to build a rigorous DP learning system, the hyperparameter tuning should be performed using private combinatorial optimization [45]. Since that is not the main focus of this paper, we skip this step for simplicity. Meanwhile, we do the following analysis of model performance

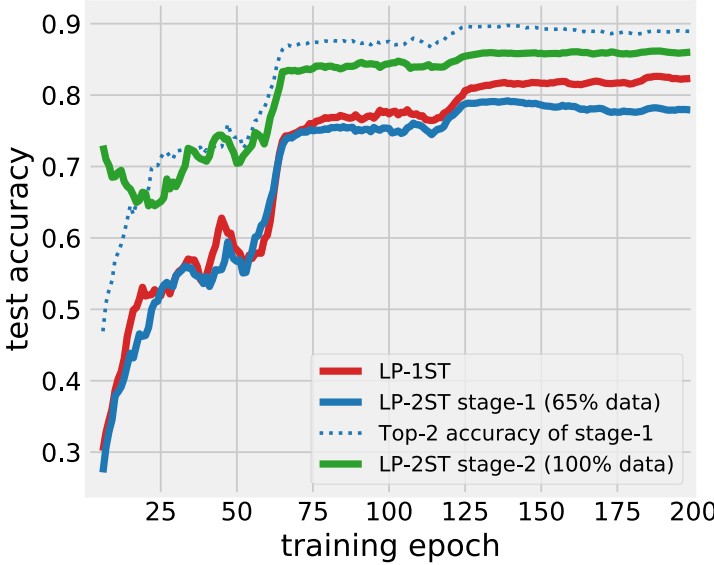

Figure 3: The learning curves of `LP-1ST` vs `LP-2ST` on CIFAR-10 ($\varepsilon = 2$).

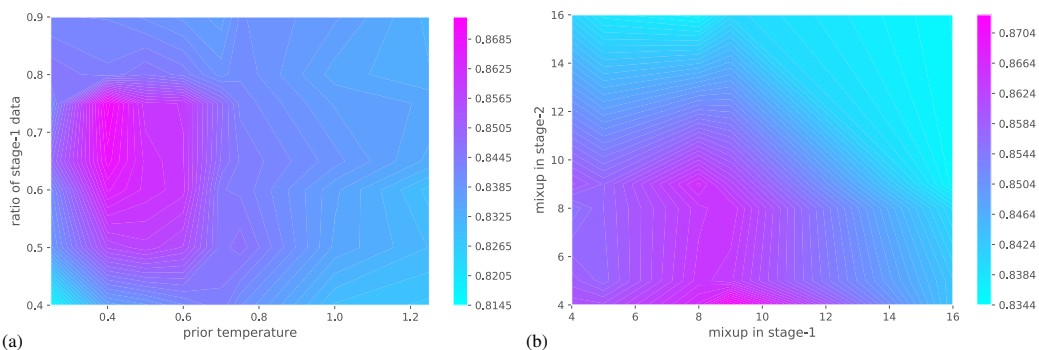

(a)                                                                      (b)

Figure 4: The final performance of `LP-2ST` on CIFAR-10 ($\varepsilon = 2$) (a) under different stage-1 / stage-2 data split and prior temperature; (b) under different mixup coefficients for stage-1 and stage-2.

under variations of different hyperparameters, which shows that the algorithms are robust in a large range of hyperparameters, and also provides some intuition for choosing the right hyperparameters.

**Data Splits and Prior Temperature.** The *data split* parameter decides the ratio of data in different stages of training. Allocating more data for stage-1 allows us to learn a better prior model for the `LP-2ST` algorithm. However, it will also decrease the number of training samples in stage-2, which reduces the utility of the learned prior model. In practice, ratios slightly higher than 50% for stage-1 strike the right balance for `LP-2ST`. We use a *temperature* parameter $t$ to modify the learned prior. Specifically, let $f_k(x)$ be the logits prediction of the learned prior model for class $k$ on input $x$. The temperature modifies the prior $\hat{p}_k(x)$ as:

$$\hat{p}_k^t(x) = \frac{\exp(f_k(x)/t)}{\sum_{k'=1}^{K} \exp(f_{k'}(x)/t)}.$$

As $t \to 0$, it sparsifies the prior by forcing it to be more confident on the top classes, and as $t \to \infty$, the prior converges to a uniform distribution. In our experiments, we find it useful to sparsify the prior, and temperatures greater than 1 are generally not helpful. Fig. 4(a) shows the performance for different combinations of data split ratio and temperature.

**Accuracy of Stage-1.** Ideally, one would want the $k$ calculated in `RRWithPrior` to satisfy the condition that the ground-truth label is always in the top-$k$ prior predictions. Because otherwise, the randomized response is *guaranteed* to be a wrong label. One way to achieve such a goal is to make

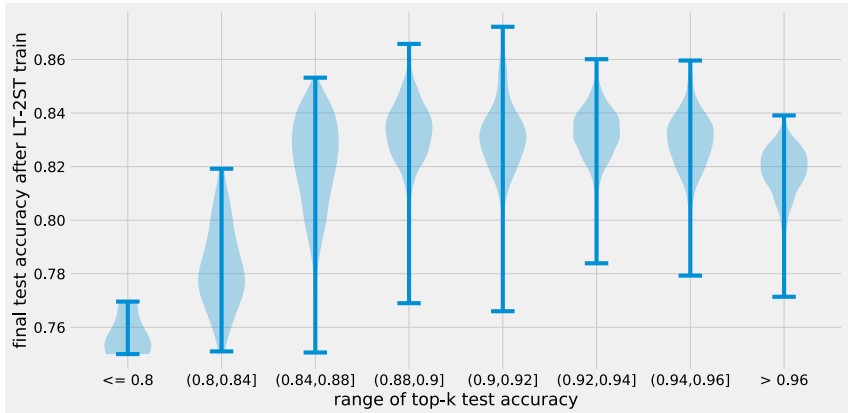

Figure 5: The relation between top-$k$ accuracy of stage-1 and the final accuracy of `LP-2ST` (CIFAR-10, $\varepsilon = 2$). The x-axis is the range of top-$k$ accuracy of stage-1 models evaluated on the test set. For each range, the violin plot shows the distribution of the final test accuracy of `LP-2ST` where the `RRWithPrior` procedure calculated an average $k$ (rounded to the nearest integer) for which the top-$k$ accuracy of the stage-1 model falls in the given range.

the stage-1 model have high top-$k$ accuracy. For example, we could allocate more data to improve the performance of stage-1 training, or tune the temperature to spread the prior to effectively increase the $k$ calculated by `RRWithPrior`. In either case, a trade-off needs to be made. In Fig. 5, we visualize the relation between top-$k$ test accuracy of stage-1 training and the final performance of `LP-2ST`. For each value range in the x-axis, we show the distribution of the final test accuracy where the average $k$ (rounded to the nearest integer) calculated in `RRWithPrior` would make the top-$k$ accuracy of the corresponding stage-1 training fall into this value range. The plot shows that the final performance drops when the top-$k$ accuracy is too low or too high. In particular, achieving near perfect top-$k$ accuracy in stage-1 is *not* desirable. Note this plot measures the top-$k$ accuracy on the *test set*, so while it is useful to observe the existence of a trade-off, it does *not* provide a procedure to choose the corresponding hyperparameters.

**Mixup Regularization.** Mixup [101] has a hyperparameter $\alpha$ that controls the strength of regularization (larger $\alpha$ corresponds to stronger regularization). We found that $\alpha$ values between 4 and 8 are generally good in our experiments, and as shown in Fig. 4(b), stage-2 typically requires less regularization than stage-1. Intuitively, this is because the data in stage-2 is less noisier than stage-1.

## F  Convex SCO with `LabelDP`

In this section, we give the proofs of the Theorem 5 and Corollary 6 for private stochastic convex optimization (SCO) and additionally prove some further, related results. We first formally introduce the setting of SCO.

Suppose we are given some *feature space* $\mathcal{X}$ (e.g., the space of all images), and *label space* $[K] = \{1, 2, \ldots, K\}$. Write $\mathcal{Z} = \mathcal{X} \times [K]$. Let $\mathcal{W} \subset \mathbb{R}^p$ be a convex *parameter space*. Let $D$ be the (Euclidean) diameter of $\mathcal{W}$, namely $D := \max_{w,w' \in \mathcal{W}} \|w - w'\|$. Suppose we are given a loss function $\ell : \mathcal{W} \times \mathcal{Z} \to \mathbb{R}$, which specifies the loss $\ell(w, z)$ for a given parameter vector $w \in \mathcal{W}$ on the example $z = (x, y)$. Given a sequence of samples $(x_1, y_1), \ldots, (x_n, y_n)$ drawn i.i.d. from a distribution $P$ over $\mathcal{Z}$, the goal is to find $w$ minimizing the popoulation risk, namely $\mathcal{L}(w, P) := \mathbb{E}_{(x,y) \sim P}[\ell(w, (x, y))]$. Write $w^\star := \arg\min_{w \in \mathcal{W}} \mathcal{L}(w, P)$. In this section, we make the following assumptions on $\ell$:

**Assumption 7** (Convexity). *For each $z \in \mathcal{Z}$, the function $w \mapsto \ell(w, z)$ is convex.*

**Assumption 8** (Lipschitzness). *For each $z \in \mathcal{Z}$, the function $w \mapsto \ell(w, z)$ is $L$-Lipschitz (with respect to the Euclidean norm).*

Under Assumptions 7 and 8, Bassily et al. [12, Theorem 4.4] showed that there is an $(\varepsilon, \delta)$-DP algorithm that given $n$ i.i.d. samples from a distribution $P$ and has access to a gradient oracle for $\ell$,

outputs some $\hat{w}$ so that the excess risk is bounded as follows:

$$\mathbb{E}[\mathcal{L}(\hat{w}, P)] - \mathcal{L}(w^\star, P) \le O\left(LD \cdot \left(\frac{\sqrt{p \log 1/\delta}}{n\varepsilon} + \frac{1}{\sqrt{n}}\right)\right). \tag{6}$$

As shown by Bassily et al. [12] (building off of previous work by Bassily et al. [10]), the rate (6) is tight up to logarithmic factors: in particular, there is a lower bound of $\Omega\left(\frac{\sqrt{p}}{n\varepsilon}\right)$ on the excess risk for any $(\varepsilon, \delta)$-DP algorithm, meaning that dimension dependence is necessary for private SCO. Subsequent work [40] showed how to obtain the rate (6) in linear (in $n$) time. We additionally remark that there has much work (e.g., [19, 59, 10, 103, 95]) on the related problem of *DP empirical risk minimization*, for which rates similar to (6), except without the $1/\sqrt{n}$ term, are attainable.

### F.1 Label-Private SGD

In this section we prove Theorem 5, showing that *dimension-independent* rates are possible in the setting of label DP privacy (in contrast to the standard setting of DP where privacy of the features must also be maintained). The algorithm that obtains the guarantee of Theorem 5 is LP-RR-SGD (Algorithm 5). Both LP-RR-SGD and the training procedure of Section 5 (which uses RRWithPrior) update the weight vectors using gradient vectors $\hat{g}_t$, which are obtained by using randomized response on the labels $y_t$ for the training examples $(x_t, y_t)$. LP-RR-SGD, however, ensures that $\hat{g}_t$ is an unbiased estimate of the true gradient, which facilitates the theoretical analysis, whereas this is not guaranteed the training procedure of Section 5.

---

**Algorithm 5 LP-RR-SGD**

---

**Input:** Distribution $P$, convex and $L$-Lipschitz loss function $\ell$, privacy parameter $\varepsilon$, convex parameter space $\mathcal{W}$, variance factor $\sigma > 0$, step size sequence $\eta_t > 0$.

1. Choose an initial weight vector $w_1 \in \mathcal{W}$.
2. For $t = 1$ to $n$:
   (a) Receive a sample $(x_t, y_t) \sim P$.
   (b) Let $\tilde{y}_t$ denote the output of RR$(y_t)$. In other words,

$$\Pr[\tilde{y}_t = \hat{y}] = \begin{cases} \frac{e^\varepsilon}{e^\varepsilon + K - 1} & \text{if } \hat{y} = y_t \\ \frac{1}{e^\varepsilon + K - 1} & \text{if } \hat{y} \ne y_t \end{cases}$$

   for all $\hat{y} \in [K]$.
   (c) Let $g_t = \nabla_w \ell(w_t, (x_t, \tilde{y}_t))$ and

$$\hat{g}_t = \frac{e^\varepsilon + K - 1}{e^\varepsilon - 1} \cdot \left(g_t - \sum_{k=1}^K \frac{\nabla_w \ell(w_t, (x_t, k))}{e^\varepsilon + K - 1}\right). \tag{7}$$

   (d) Let $w_{t+1} \leftarrow \Pi_\mathcal{W}(w_t - \eta_t \cdot \hat{g}_t)$.
3. Output $\hat{w} := w_{n+1}$.

---

We now restate Theorem 5 formally below:

**Theorem 9** (Formal version of Theorem 5). *For any $\varepsilon \in (0, 1)$, the algorithm* LP-RR-SGD *satisfies the requirement of $\varepsilon$-LabelDP; moreover, if run with step size $\eta_t = \frac{D\varepsilon}{6KL\sqrt{t}}$, its output $\hat{w}$ satisfies*

$$\mathbb{E}[\mathcal{L}(\hat{w}, P)] - \mathcal{L}(w^\star, P) \le O\left(\frac{DLK \log(n)}{\varepsilon \sqrt{n}}\right).$$

We remark that even in the non-private setting, a lower bound of $\Omega(DL/\sqrt{n})$ is known on the excess risk for stochastic convex optimization [73, 6], meaning that Theorem 9 is tight up to a factor of $O(K \log n / \varepsilon)$. In Section F.3, we improve the lower bound to $\tilde{\Omega}(DL/\sqrt{\varepsilon n})$ for small $\varepsilon \le 1$ (where $\tilde{\Omega}$ hides a logarithmic factor in $1/\varepsilon$). Hence, our bound above is tight to within a factor of $\tilde{O}(K \log n / \sqrt{\varepsilon})$.

*Proof of Theorem 9.* We first verify the privacy property of LP-RR-SGD. For any two points $(x_t, y_t), (x_t, y'_t)$, differing only in their label, if we let $\hat{g}_t, \hat{g}'_t$ be the vectors defined in (7) for each

of these points, respectively, then it is immediate from definition of $Q_t$ that for any subset $\mathcal{S} \subset \mathbb{R}^p$, $\frac{\Pr[\hat{g}_t \in \mathcal{S}]}{\Pr[\hat{g}'_t \in \mathcal{S}]} \leq e^\varepsilon$. That LP-RR-SGD is $\varepsilon$-LabelDP follows immediately from the post-processing property of DP.

Next we establish the uility guarantee. Note that by definition of $\hat{g}_t$, we have that

$$\mathbb{E}_{\tilde{y}_t}[\hat{g}_t] = \frac{e^\varepsilon + K - 1}{e^\varepsilon - 1} \cdot \left( \frac{e^\varepsilon \cdot \nabla_w \ell(w_t, (x_t, y_t))}{e^\varepsilon + K - 1} + \sum_{k \neq y_t} \frac{\nabla_w \ell(w_t, (x_t, k))}{e^\varepsilon + K - 1} - \sum_{k=1}^{K} \frac{\nabla_w \ell(w_t, (x_t, k))}{e^\varepsilon + K - 1} \right)$$
$$= \nabla_w \ell(w_t, (x_t, y_t)),$$

i.e., $\hat{g}_t$ is an unbiased estimate of $\nabla_w \ell(w_t, (x_t, y_t))$.

Next, we bound the variance of the gradient error $\hat{g}_t - \nabla_w \ell(w_t, (x_t, y_t))$, as follows:

$$\mathbb{E}_{\tilde{y}_t} \left[ \|\hat{g}_t - \nabla_w \ell(w_t, (x_t, y_t))\|^2 \right]$$
$$\leq 2 \left( \frac{2K}{\varepsilon} \right)^2 \cdot \mathbb{E}_{\tilde{y}_t} \left[ \left\| \left( g_t - \sum_{k=1}^{K} \frac{\nabla_w \ell(w_t, (x_t, k))}{e^\varepsilon + K - 1} \right) \right\|^2 \right] + 2 \|\nabla_w \ell(w_t, (x_t, y_t))\|^2$$
$$\leq \frac{32K^2 L^2}{\varepsilon^2} + 2L^2 \leq \frac{36K^2 L^2}{\varepsilon^2},$$

where we have used that $\ell$ is $L$-Lipschitz, $\varepsilon \leq 1$, and that $K \geq 2$.

Using Shamir and Zhang [81, Theorem 2] with gradient moment $G^2 := \frac{36K^2 L^2}{\varepsilon^2}$, we get that for step size choices $\eta_t := \frac{D}{G\sqrt{t}}$, the output $\hat{w}$ of LP-RR-SGD satisfies

$$\mathbb{E}[\mathcal{L}(\hat{w}, P)] - \mathcal{L}(w^\star, P) \leq O\left( \frac{DG \log n}{\sqrt{n}} \right) \leq O\left( \frac{DLK \log(n)}{\varepsilon \sqrt{n}} \right). \qquad \square$$

Now we prove Corollary 6; a formal version of the corollary is stated below.

**Corollary 10** (Formal version of Corollary 6). *Suppose that we are given a prior $\mathbf{p}^x$ for every $x$ and let $Y_k^x$ denote the set of top-$k$ labels with respect to $\mathbf{p}^x$. Then, for any $\varepsilon \in (0, 1)$, there is an $\varepsilon$-LabelDP algorithm which outputs $\hat{w} \in \mathcal{W}$ satisfying*

$$\mathbb{E}[\mathcal{L}(\hat{w}, P)] - \min_w \mathcal{L}(w, P) \leq O\left( DL \cdot \left( \frac{k \log n}{\varepsilon \sqrt{n}} + \Pr_{(x,y) \sim P}[y \notin Y_k^x] \right) \right) \qquad (8)$$

*Proof.* Suppose we are given access to samples $(x, y)$ drawn from a distribution $P$ on $\mathcal{X} \times [K]$. For a pair $(x, y) \in \mathcal{X} \times [K]$, define a random pair $\xi((x, y)) \in \mathcal{X} \times [K]$, by setting $\xi((x, y)) = (x, y)$ if $y \in Y_k^x$, and otherwise letting $\xi((x, y))$ to be drawn uniformly over the set $\{(x, k') : k' \in Y_k^x\}$. Let $P'$ be the distribution of $\xi((x, y))$, where $(x, y) \sim P$. For any $w_1, w_2 \in \mathcal{W}$, it follows that

$$|(\mathcal{L}(w_1, P) - \mathcal{L}(w_2, P)) - (\mathcal{L}(w_1, P') - \mathcal{L}(w_2, P'))|$$
$$= \left| \int_{\mathcal{Z}} [\ell(w_1, (x, y)) - \ell(w_2, (x, y))] dP((x, y)) - \int_{\mathcal{Z}} [\ell(w_1, (x, y)) - \ell(w_2, (x, y))] dP'((x, y)) \right|$$
$$\leq \left| \int_{\{(x,y) : y \notin Y_k^x\}} ([\ell(w_1, (x, y)) - \ell(w_2, (x, y))] - [\ell(w_1, \xi((x, y))) - \ell(w_2, \xi((x, y)))]) dP((x, y)) \right|$$
$$\leq 2DL \cdot \Pr_{(x,y) \sim P}[y \notin Y_k^x], \qquad (9)$$

where the last step uses that $|\ell(w_1, (x, y)) - \ell(w_2, (x, y))| \leq L\|w_1 - w_2\| \leq LD$ for all $w_1, w_2 \in \mathcal{W}$. Now we simply run the algorithm LP-RR-SGD, except that when we receive a point $(x, y) \sim P$, we pass the example $\xi((x, y))$ to LP-RR-SGD (instead of $(x, y)$), and we let the set of possible labels be $Y_k^x$ (instead of $[K]$). Since each such example $\xi((x, y))$ is only passed to LP-RR-SGD once, the resulting allgorithm is still $\varepsilon$-LabelDP. Since the label of $\xi((x, y))$ belongs to $Y_k^x$, which has size $k$ for all $x$, Theorem 9 gives that the output $\hat{w}$ of LP-RR-SGD satisfies $\mathbb{E}[\mathcal{L}(\hat{w}, P')] - \min_w \mathcal{L}(w, P') \leq$

$O\left(\frac{DLk\log(n)}{\varepsilon\sqrt{n}}\right)$. Next (9) gives that, for any fixed $\hat{w}$, letting $w_{P'}^{\star} = \arg\min_w \mathcal{L}(w, P')$, $w_P^{\star} = \arg\min_w \mathcal{L}(w, P)$,

$$
\begin{aligned}
\mathcal{L}(\hat{w}, P) - \mathcal{L}(w_P^{\star}, P) &\leq \mathcal{L}(\hat{w}, P') - \mathcal{L}(w_P^{\star}, P') + 2DL \cdot \Pr_{(x,y)\sim P}[y \notin Y_k^x] \\
&\leq \mathcal{L}(\hat{w}, P') - \mathcal{L}(w_{P'}^{\star}, P') + 2DL \cdot \Pr_{(x,y)\sim P}[y \notin Y_k^x], \quad (10)
\end{aligned}
$$

where (10) follows since $\mathcal{L}(w_{P'}^{\star}, P') \leq \mathcal{L}(w_P^{\star}, P')$ by definition of $w_{P'}^{\star}$. (8) is an immediate consequence. $\qquad\square$

### F.2 A Better Bound for Approximate DP

Next we introduce an algorithm, LP-Normal-SGD (Algorithm 6), which shows how to improve upon the excess risk bound of Theorem 9 by a factor of $\sqrt{K}$, if we relax the privacy requirement to approximate LabelDP (i.e., $(\varepsilon, \delta)$-LabelDP with $\delta > 0$). LP-SGD performs a single pass of SGD over the input dataset, with the following modification: it adds a Gaussian noise vector to each gradient vector with nonzero variance only in the $K$-dimensional subspace $\mathcal{L}_t$ corresponding to the $K$ possible labels for each point $x_t$. This means that the norm of a typical noise vector scales only as $\sqrt{K}$ as opposed to the scaling $\sqrt{p}$, which similar algorithms for the standard setting of DP (e.g., [10]) obtain.

---

**Algorithm 6 LP-Normal-SGD**

---

**Input:** Distribution $P$ over $\mathcal{X} \times [K]$, convex and $L$-Lipschitz loss function $\ell$, privacy parameters $\varepsilon, \delta$, convex parameter space $\mathcal{W}$, variance factor $\sigma > 0$, step size sequence $\eta_t > 0$.

1. Choose an initial weight vector $w_1 \in \mathcal{W}$.
2. For $t = 1$ to $n$:
   (a) Receive $(x_t, y_t) \sim P$.
   (b) Let $\tilde{b}_t \sim \mathcal{N}(0, \sigma^2 I_p)$.
   (c) Let $\mathcal{L}_t \leftarrow \text{span}\{\nabla_w \ell(w_t, (x_t, k)) : k \in [K]\} \subset \mathbb{R}^p$.
   (d) Let $b_t \leftarrow \Pi_{\mathcal{L}_t}(\tilde{b}_t)$ denote the Euclidean projection of $\tilde{b}_t$ onto $\mathcal{L}_t$.
   (e) Let $w_{t+1} \leftarrow \Pi_{\mathcal{W}}(w_t - \eta_t \cdot (\nabla_w \ell(w_t, (x_t, y_t)) + b_t))$.
3. Output $\hat{w} := w_{n+1}$.

---

**Proposition 11.** *There is a constant $C > 0$ so that the following holds. For any $\varepsilon, \delta \in (0, 1)$, $\sigma = \frac{CL\sqrt{\log 1/\delta}}{\varepsilon}$, $\eta_t = \frac{D}{\sqrt{(L^2 + K\sigma^2) \cdot t}}$, the algorithm LP-SGD (Algorithm 6) is $(\varepsilon, \delta)$-LabelDP and satisfies the following excess risk bound:*

$$
\mathbb{E}[\mathcal{L}(\hat{w}, S)] - \mathcal{L}(w^{\star}, S) \leq O\left(\frac{DL\sqrt{K\log 1/\delta} \cdot \log(n)}{\varepsilon\sqrt{n}}\right).
$$

*Proof of Proposition 11.* We first argue that the privacy guarantee holds. Note that for any $k, k' \in [n]$, for any $x \in \mathcal{X}, w \in \mathcal{W}$, we have $\|\nabla_w \ell(w, (x, k)) - \nabla_w \ell(w, (x, k'))\| \leq 2L$. Therefore, for any $w_t \in \mathcal{W}$, the mechanism

$$
k \mapsto \nabla_w \ell(w_t, (x_{i_t}, k)) + b_t
$$

is $(\varepsilon, \delta)$-DP as long as $\sigma \geq \frac{CL\sqrt{\log 1/\delta}}{\varepsilon}$, for some constant $C > 0$ [32]. Since each $(x_t, y_t)$ is used in only a single iteration of LP-Normal-SGD, it follows from the post-processing of DP that LP-Normal-SGDis $(\varepsilon, \delta)$-LabelDP for this choice of $\sigma$.

Next we establish the utility guarantee. Since, for each $t \in [n]$, $\mathcal{L}_t$ is a subspace of $\mathbb{R}^p$ of at most $K$ dimensions, it holds that for each $t$, $\mathbb{E}[\|b_t\|^2] \leq K\sigma^2$. Thus $\mathbb{E}\left[\|\nabla_w \ell(w_t, (x_{i_t}, y_{i_t})) + b_t\|^2\right] \leq L^2 + K\sigma^2$. Using Shamir and Zhang [81, Theorem 2] with gradient moment $G^2 := L^2 + K\sigma^2$, we get that for step size choices $\eta_t := \frac{D}{G\sqrt{t}}$, it holds that

$$
\mathbb{E}[\mathcal{L}(\hat{w}, S)] - \mathcal{L}(w^{\star}, S) \leq O\left(\frac{DG\log n}{\sqrt{n}}\right) \leq O\left(\frac{DL\sqrt{K\log 1/\delta} \cdot \log(n)}{\varepsilon\sqrt{n}}\right). \qquad\square
$$

## F.3  Lower Bound on Population Risk

In this section, we prove the following lower bound on excess risk, which is tight with respect to (11) in Proposition 11 up to a factor of $\tilde{O}(\sqrt{K/\varepsilon})$.

**Proposition 12.** *For any $\varepsilon \in (0, 1], D, L > 0$ and any sufficiently large $n \in \mathbb{N}$ and sufficiently small $\delta > 0$ (both depending on $\varepsilon$), the following holds: for any $(\varepsilon, \delta)$-LabelDP algorithm A, there exists a loss function $\ell$ that is L-Lipschitz and convex, and a distribution $P$ for which*

$$\mathbb{E}_{\tilde{S} \sim P^{\otimes n}, \hat{w} \sim \mathsf{A}(\tilde{S})}[\mathcal{L}(\hat{w}, P)] - \mathcal{L}(w^{\star}, P) \geq \tilde{\Omega}\left(\frac{DL}{\sqrt{\varepsilon n}}\right). \tag{11}$$

We remark that the lower bound of $\Omega(DL/\sqrt{n})$ is well known for *non-private* SCO. This lower bound applies to our setting as well and thus the lower bound in Proposition 12 can be viewed as an improvement of a factor for $\tilde{\Omega}(1/\sqrt{\varepsilon})$ over the non-private lower bound.

We prove Equation (11) by first proving an analogous bound in the empirical loss minimization (ERM) setting and then deriving SCO via a known reduction.

## F.4  Lower Bound on Excess Risk for ERM

Recall that in ERM setting, we are given a set $S = \{(x_1, y_1), \ldots, (x_n, y_n)\} \subseteq \mathcal{Z}$ of $n$ labelled examples. The empirical risk of $w$ is defined as $\mathcal{L}(w, S) := \frac{1}{n} \sum_{i=1}^{n} \ell(w, (x, y))$. Here we would like to devise an algorithm that minimizes the excess empirical risk, i.e., $\mathbb{E}[\mathcal{L}(\hat{w}, S)] - \mathcal{L}(w^{\star}, S)$ where $\hat{w}$ is the output of the algorithm and $w^{\star} := \arg\min_{w \in \mathcal{W}} \mathcal{L}(w, S)$.

We start by proving the following lower bound on excess risk for LabelDP ERM algorithms. Note that the lower bound does not yet grow as $\varepsilon$ decreases; that version of the lower bound will be proved later in this section.

**Proposition 13.** *For any $\varepsilon, D, L, \delta > 0, K \geq 2$ and $n \in \mathbb{N}$ such that $\varepsilon \leq O(1), \delta \leq 1 - \Omega(1)$, the following holds: for any $(\varepsilon, \delta)$-LabelDP algorithm A, there exists a loss function $\ell$ that is L-Lipschitz and convex, and a dataset $\tilde{S}$ of size $n$ for which*

$$\mathbb{E}_{\hat{w} \sim \mathsf{A}(\tilde{S})}[\mathcal{L}(\hat{w}, \tilde{S})] - \mathcal{L}(w^{\star}, \tilde{S}) \geq \Omega\left(\frac{DL}{\sqrt{n}}\right). \tag{12}$$

*Proof.* Let $\mathcal{W} := \{w \in \mathbb{R}^d : \|\mathbf{w}\| \leq D/2\}$ and $\mathcal{X} := \{x \in \mathbb{R}^d : \|x\| \leq 1\}$. We define the loss to be

$$\ell(w, (x, y)) := \begin{cases} L \cdot \langle w, x \rangle & \text{if } y = 1, \\ -L \cdot \langle w, x \rangle & \text{if } y = 2, \\ 0 & \text{otherwise.} \end{cases}$$

Note that the diameter of $\mathcal{W}$ is $D$ and $\ell(\cdot, (x, y))$ is convex and L-Lipschitz. Consider any $(\varepsilon, \delta)$-LabelDP algorithm A. Let $e_i \in \mathbb{R}^n$ be the $i$th standard basis vector. Consider a dataset $S = \{(e_1, y_1), \ldots, (e_n, y_n)\}$ where $y_1, \ldots, y_n \in \{1, 2\}$ are random labels which are 1 w.p. 0.5 and 2 otherwise. For notational convenience, we write $\tilde{y}_i$ to denote $2y_i - 3 \in \{-1, 1\}$. By the $(\varepsilon, \delta)$-LabelDP guarantee of A, we have

$$\Pr_{S, \hat{w} \sim \mathsf{A}(S)}[\tilde{y}_i \cdot \langle \hat{w}, e_i \rangle > 0]$$

$$= \frac{1}{2} \Pr_{S, \hat{w} \sim \mathsf{A}(S)}[\langle \hat{w}, e_i \rangle < 0 \mid \tilde{y}_i = -1]$$

$$\quad + \frac{1}{2} \Pr_{S, \hat{w} \sim \mathsf{A}(S)}[\langle \hat{w}, e_i \rangle > 0 \mid \tilde{y}_i = 1]$$

$$\leq \frac{1}{2} \cdot \left(e^{\varepsilon} \cdot \Pr_{S, \hat{w} \sim \mathsf{A}(S)}[\langle \hat{w}, e_i \rangle < 0 \mid \tilde{y}_i = 1] + \delta\right)$$

$$\quad + \frac{1}{2}\left(e^{\varepsilon} \cdot \Pr_{S, \hat{w} \sim \mathsf{A}(S)}[\langle \hat{w}, e_i \rangle > 0 \mid \tilde{y}_i = -1] + \delta\right)$$

$$= e^{\varepsilon} \cdot \Pr_{S, \hat{w} \sim \mathsf{A}(S)}[\tilde{y}_i \cdot \langle \hat{w}, e_i \rangle < 0] + \delta.$$

This implies that

$$\Pr_{S,\hat{w}\sim A(S)}[\tilde{y}_i \cdot \langle \hat{w}, e_i \rangle > 0] \leq \frac{e^\varepsilon + \delta}{e^\varepsilon + 1}. \tag{13}$$

Letting $I_{\hat{w},S} := \{i \in [n] : \tilde{y}_i \cdot \langle \hat{w}, e_i \rangle > 0\}$ for any $S, \hat{w}$,

$$\mathbb{E}_{S,\hat{w}\sim A(S)}[|I_{\hat{w},S}|] = \sum_{i \in [n]} \Pr_{S,\hat{w}\sim A(S)}[\tilde{y}_i \cdot \langle \hat{w}, e_i \rangle > 0]$$

$$\overset{(13)}{\leq} \left(\frac{e^\varepsilon + \delta}{e^\varepsilon + 1}\right) n. \tag{14}$$

Consider any $S$ as generated above; it is obvious to see that $w^\star = \frac{D}{2} \cdot \left(\frac{1}{\sqrt{n}} \sum_{i \in [n]} \tilde{y}_i e_i\right)$, which results in $\mathcal{L}(w^\star, S) = -\frac{DL}{2\sqrt{n}}$. On the other hand, for any $\hat{w}$,

$$\mathcal{L}(\hat{w}, S) = \frac{1}{n} \sum_{i \in [n]} \ell(\hat{w}, (e_i, y_i)) = \frac{1}{n} \sum_{i \in [n]} -L \langle \hat{w}, \tilde{y}_i \cdot e_i \rangle$$

$$\geq \frac{1}{n} \sum_{i \in I_{\hat{w},S}} -L \langle \hat{w}, \tilde{y}_i \cdot e_i \rangle = \frac{-L}{n} \left\langle \hat{w}, \sum_{i \in I_{\hat{w},S}} \tilde{y}_i \cdot e_i \right\rangle$$

$$\geq \frac{-L}{n} \cdot \|\hat{w}\| \cdot \left\| \sum_{i \in I_{\hat{w},S}} \tilde{y}_i \cdot e_i \right\| \geq \frac{-L}{n} \cdot \frac{D}{2} \cdot \sqrt{|I_{\hat{w},S}|}, \tag{15}$$

where we used Cauchy–Schwarz inequality in the second inequality above. As a result, we have

$$\mathbb{E}_S[\mathbb{E}_{\hat{w}\sim A(S)}[\mathcal{L}(\hat{w}, S)] - \mathcal{L}(w^\star, S)]$$

$$= \mathbb{E}_{S,\hat{w}\sim A(S)}[\mathcal{L}(\hat{w}, S)] + \frac{DL}{2\sqrt{n}}$$

$$\overset{(15)}{\geq} \frac{-DL}{2n} \cdot \mathbb{E}_{S,\hat{w}\sim A(S)}\left[\sqrt{|I_{\hat{w},S}|}\right] + \frac{DL}{2\sqrt{n}}$$

$$\geq \frac{-DL}{2n} \cdot \sqrt{\mathbb{E}_{S,\hat{w}\sim A(S)}[|I_{\hat{w},S}|]} + \frac{DL}{2\sqrt{n}}$$

$$\overset{(14)}{\geq} \frac{DL}{2\sqrt{n}} \left(-\sqrt{\frac{e^\varepsilon + \delta}{e^\varepsilon + 1}} + 1\right)$$

$$\geq \Omega(DL/\sqrt{n}),$$

where the second inequality follows from Cauchy–Schwarz inequality and the last inequality follows from our assumption that $\delta \leq 1 - \Omega(1)$ and $\varepsilon \leq O(1)$. $\qquad\square$

To make the lower bound above grows with $1/\sqrt{\varepsilon}$ for $\varepsilon \leq 1$, we will apply the technique used in [89]. Recall that a pair of datasets are said to be $k$-neighbor if they differ in at most $k$ labels. The following is a well-known bound, so-called *group privacy*; see e.g. Steinke and Ullman [89, Fact 2.3]. (Typically this fact is stated for the standard DP but it applies to LabelDP in the same manner.)

**Fact 14.** *Let A be any $(\varepsilon, \delta)$-LabelDP algorithm. Then, for any $k$-neighboring database $S, S'$ and every subset $T$ of the output, we have $\Pr[A(S) \subseteq T] \leq e^{k\varepsilon} \cdot \Pr[A(S') \subseteq T] + \frac{e^{k\varepsilon}-1}{e^\varepsilon-1} \cdot \delta$.*

We can now prove the following lower bound that grows with $1/\sqrt{\varepsilon}$ by simplying replicating each element $1/\varepsilon$ times.

**Lemma 15.** *For any $\varepsilon' \in (0,1], D, L, \delta' > 0, K \geq 2$ and $n \in \mathbb{N}$ such that $n \geq 1/\gamma, \delta' \leq \Omega(\varepsilon')$, the following holds: for any $(\varepsilon', \delta')$-LabelDP algorithm $A'$, there exists a loss function $\ell$ that is $L$-Lipschitz and convex, and a dataset $\tilde{S}'$ of size $n$ for which*

$$\mathbb{E}_{\hat{w}\sim A'(\tilde{S}')}[\mathcal{L}(\hat{w}, \tilde{S}')] - \mathcal{L}(w^\star, \tilde{S}') \geq \Omega\left(\frac{DL}{\sqrt{\varepsilon'}n}\right). \tag{16}$$

*Proof.* Suppose for the sake of contradiction there exists $(\varepsilon, \delta)$-LabelDP algorithm $A'$ such that $\mathbb{E}_{\hat{w} \sim A'(\tilde{S}')}[\mathcal{L}(\hat{w}, \tilde{S}')] - \mathcal{L}(w^\star, \tilde{S}') \leq o\left(\frac{DL}{\sqrt{\varepsilon'}n}\right)$. Let $k = \lfloor 1/\varepsilon \rfloor$. We construct an algorithm $A$ as follows: on input $\tilde{S}$, it replicates each element of $\tilde{S}$ $k$ times to construct a dataset $\tilde{S}'$. It then returns $A'(\tilde{S}')$. From the utility guarantee of $A'$, we have $\mathbb{E}_{\hat{w} \sim A(\tilde{S})}[\mathcal{L}(\hat{w}, \tilde{S})] - \mathcal{L}(w^\star, \tilde{S}) \leq o\left(\frac{DL}{\sqrt{n}}\right)$. Furthermore, Fact 14 ensures that $A$ is $(\varepsilon, \delta)$-DP for $\varepsilon = k\varepsilon' \leq 1$ and $\delta = \frac{e^{k\varepsilon'}-1}{e^{\varepsilon'}-1}\delta' \leq O(\delta'/\varepsilon')$. When $\delta' = C/\varepsilon'$ for any sufficiently small $C > 0$, $A$ violates Proposition 13, concluding our proof. $\qquad\square$

### F.5 From ERM to SCO

Bassily et al. [12][6] gave a reduction from private SCO to private ERM. Although this bound is proved in the context of standard (both label and sample) DP, it is not hard to see that a similar bound holds for LabelDP with exactly the same proof. To summarize, their proof yields the following bound:

**Lemma 16.** *For any $\gamma, \varepsilon > 0$ and $\delta \in (0, 1/2)$, suppose that there is an $\left(\frac{\varepsilon}{4\log(2/\delta)}, \frac{e^{-\varepsilon}\delta}{8\log(2/\delta)}\right)$-LabelDP algorithm that yields expected excess population risk of for SCO is at most $\gamma$. Then, there exists an $(\varepsilon, \delta)$-LabelDP algorithm for convex ERM (with the same parameters $D, L, n$) with excess empirical risk at most $\gamma$.*

Plugging this into Lemma 15, we arrive at Proposition 12.

## G  Generalization Bounds for RR with Prior

Let $\mathcal{X}, \mathcal{Z}$ be similar to the previous section and $\mathcal{Y} = [K]$ be the class of labels. We consider a setting where there is a concept class $\mathcal{F}$ of functions $f : \mathcal{X} \to \mathbb{R}$. Given $n$ samples drawn i.i.d. from some distribution $P$ on $\mathcal{Z}$, we would like to output a function $f$ with a small *population risk*, which is defined as $\mathcal{L}(f; P) = \mathbb{E}_{(x,y)\sim P}[\ell(f(x), (x,y))]$., where $\ell : \mathbb{R} \times \mathcal{Z} \to [0, 1]$ is a loss function. Throughout this section, we assume that $\ell$ is $L$-Lipschitz (Assumption 8).

**Priors and Randomized Response.** Let $k \leq K$ be a positive integer. We work in the same setting as Corollary 10, i.e. we assume a prior $\mathbf{p}^x$ for every $x$ and let $Y_k^x$ denote the set of top-$k$ labels with respect to $\mathbf{p}^x$. We let $\tilde{P}$ be the distribution where we first draw $(x, y) \sim P$ and then output $(x, \tilde{y})$ where $\tilde{y} \sim \text{RRTop-}k_{\mathbf{p}^x}(y)$ with DP parameter $\varepsilon$.

**Debiased Loss Function.** Let $p_{k,\varepsilon}$ denote $\frac{1}{e^\varepsilon+k-1}$. We consider a debiased version of the loss $\ell$; this was done before in [72] for the case of binary classification with noisy labels. In our setting, it generalizes to the following definition:

$$\tilde{\ell}(t, (x, y)) := \frac{1}{1 - k \cdot p_{k,\varepsilon}} \cdot \left(\ell(t, (x, y)) - \sum_{y' \in Y_k^x} p_{k,\varepsilon} \cdot \ell(t, (x, y'))\right). \tag{17}$$

For a set $S$ of $n$ labeled examples $(x_1, y_1), \ldots, (x_n, y_n) \in \mathcal{Z}$, its *empirical risk* (w.r.t loss $\tilde{\ell}$) as $\tilde{\mathcal{L}}(f; S) = \frac{1}{n}\sum_{i=1}^n \tilde{\ell}(f(x_i), (x_i, y_i))$.

We consider simple $\varepsilon$-LabelDP algorithm that randomly draws $n$ i.i.d. samples $S$ from $P$, apply $(\varepsilon$-LabelDP) RRTop-$k$ on each of the label to get a randomized dataset $\tilde{S}$, and finally apply empirical risk minimization w.r.t. the debiased loss function $\tilde{\ell}$ on $\tilde{S}$. We remark that this algorithm is exactly the same as drawing $n$ samples i.i.d. from $\tilde{P}$ and apply empirical risk minimization (again w.r.t. $\tilde{\ell}$). Our main result of this section is a generalization bound roughly saying that the empirical risk (w.r.t. $\tilde{\ell}$) is small iff the popultion risk (w.r.t. $\ell$) is small. This is stated more formally below, where $\mathcal{R}_{n,D}(\mathcal{F})$ denote the Rademacher Complexity of $\mathcal{F}$ (defined below in Definition G.2).

**Theorem 17.** *Let $P_\mathcal{X}$ be the marginal of $P$ over $\mathcal{X}$. Let $\tilde{S}$ be a set of $n$ i.i.d. labeled samples drawn from $\tilde{P}$. Then, with probability at least $1 - \beta$, the following holds for all $f \in \mathcal{F}$:*

$$|\tilde{\mathcal{L}}(f; \tilde{S}) - \mathcal{L}(f; P)| \leq 2L \cdot \frac{1 + k \cdot p_{k,\varepsilon}}{1 - k \cdot p_{k,\varepsilon}} \cdot \mathcal{R}_{n,P_\mathcal{X}}(\mathcal{F}) + \sqrt{\frac{\log(2/\beta)}{2n}} + \Pr_{(x,y)\sim P}[y \notin Y_k^x]. \tag{18}$$

---

[6]See the proof in Appendix D of the arXiv version of their paper [11].

Via standard techniques (see e.g. [72]), the above bound imply that the empirical risk minimizer incurs excess loss similar to the bound in Equation (18) (within a factor of 2).

Recall that RRTop-$k$ can of course be thought of RRWithPrior in the case when e.g. the prior $\mathbf{p}^x$ is uniform over the $k$ labels in $Y_k^x$. Thus, Theorem 17 can be viewed as a generalization bound for RRWithPrior with these "uniform top-$k$" priors.

## G.1 Additional Preliminaries

To prove Theorem 17, we need several additional observations and definitions. In addition to the previously defined $\mathcal{L}(f; P), \tilde{\mathcal{L}}(f; S)$, we analogously use $\tilde{\mathcal{L}}(f; P), \mathcal{L}(f; S)$ to denote the population risk w.r.t. $\tilde{\ell}$ on distribution $P$ and the empirical risk w.r.t. $\ell$ on the labeled sample set $S$ respectively.

**Properties of the Debiased Loss Function.** We will start by proving a few basic properties of the debiased loss functions. The first two lemmas are simple to check:

**Lemma 18.** *If $y \in Y_k^x$, it holds that $\mathbb{E}_{\tilde{y} \sim \text{RRTop-}k_{\mathbf{p}^x}(y)}[\tilde{\ell}(t, (x, \tilde{y}))] = \ell(t, (x, y))$.*

**Lemma 19.** *$\tilde{\ell}$ is $L \cdot \frac{1 + k \cdot p_{k,\varepsilon}}{1 - k \cdot p_{k,\varepsilon}}$-Lipschitz (in $t$ for every fixed $x, y$).*

Finally, we observe that the population risk w.r.t. $\tilde{\ell}$ on distribution $\tilde{P}$ is close to that w.r.t. $\ell$ on $P$:

**Lemma 20.** *For any function $f$, we have*

$$|\mathcal{L}(f; P) - \tilde{\mathcal{L}}(f, \tilde{P})| \leq \Pr_{(x,y) \sim P}[y \notin Y_k^x]. \tag{19}$$

*Proof.* We can write

$$|\mathcal{L}(f; P) - \tilde{\mathcal{L}}(f; \tilde{P})| = |\mathbb{E}_{(x,y) \sim P}[\ell(f, (x, y))] - \mathbb{E}_{(x,y) \sim P, \tilde{y} \sim \text{RRTop-}k_{\mathbf{p}^x}(y)}[\ell(f, (x, \tilde{y}))]|$$
$$\leq \mathbb{E}_{(x,y) \sim P}[|\ell(f, (x, y)) - \mathbb{E}_{\tilde{y} \sim \text{RRTop-}k_{\mathbf{p}^x}(y)}[\ell(f, (x, \tilde{y}))]|].$$

Due to Lemma 18, the inner term is zero whenever $y \in Y_k^x$; furthermore, since the range of $\ell$ is in $[0, 1]$, the last term is at most $\Pr_{(x,y) \sim P}[y \notin Y_k^x]$ as desired. $\square$

**Rademacher Complexity.** Given a space $\mathcal{V}$ and a distribution $D$ over $\mathcal{V}$, we let $S$ be a set of examples $v_1, \ldots, v_n$ drawn i.i.d. from $D$. We also let $\mathcal{F}$ be a class of functions $f : \mathcal{V} \to \mathbb{R}$.

**Definition G.1** (Empirical Rademacher Complexity). The *empirical Rademacher complexity* of $\mathcal{F}$ is defined as:

$$\hat{\mathcal{R}}_{n,S}(\mathcal{F}) = \mathbb{E}_{\sigma_1, \ldots, \sigma_n} \left[ \sup_{f \in \mathcal{F}} \left( \frac{1}{n} \sum_{i=1}^{n} \sigma_i f(v_i) \right) \right], \tag{20}$$

where $\sigma_1, \ldots, \sigma_n$ are i.i.d. random variables sampled uniformly at random from $\{\pm 1\}$.

**Definition G.2** (Rademacher Complexity). The Rademacher complexity of $\mathcal{F}$ is defined as

$$\mathcal{R}_{n,D}(\mathcal{F}) = \mathbb{E}[\hat{\mathcal{R}}_{n,S}(\mathcal{F})], \tag{21}$$

where the expectation is over the randomness of the subset $S$ which consists of $n$ elements chosen i.i.d. from $D$.

We also need the following two known lemmas.

**Lemma 21** ([14]). *Let $D$ be a distribution and $\beta \in (0, 1)$. If $\mathcal{F} \subseteq \{f : \mathcal{V} \to [0, 1]\}$ and $S = \{v_1, \ldots, v_n\}$ consists of $n$ elements drawn i.i.d. from $D$, then with probability at least $1 - \beta$ over the randomness of $S$, for every function $f \in \mathcal{F}$, it holds that*

$$\left| \mathbb{E}_{v \sim D}[f(v)] - \frac{1}{n} \sum_{i=1}^{n} f(v_i) \right| \leq 2\mathcal{R}_{n,D}(\mathcal{F}) + \sqrt{\frac{\ln(2/\beta)}{n}}. \tag{22}$$

The following lemma is a standard bound for the empirical Rademacher complexity (and follows from the Ledoux-Talagrand contraction inequality [62]).

**Lemma 22.** *Let $\mathcal{F} \subseteq \{f : \mathcal{X} \to \mathbb{R}\}$. Let $S$ be a multiset of $n$ (possibly repeated) elements $v_1, \ldots, v_n \in \mathcal{X}$. Moreover, let $\Phi_1, \ldots, \Phi_n$ be L-Lipschitz functions mapping $\mathbb{R}$ to $\mathbb{R}$. Then, it holds that*

$$\mathbb{E}_{\sigma_1,\ldots,\sigma_n}\left[\sup_{f \in \mathcal{F}}\left(\frac{1}{n}\sum_{i=1}^{n}\sigma_i\Phi_i(f(v_i))\right)\right] \le L \cdot \hat{\mathcal{R}}_{n,S}(\mathcal{F}). \tag{23}$$

## G.2 Proof of Theorem 17

With the preliminaries ready, we can now prove Theorem 17.

*Proof of Theorem 17.* With probability $1 - \beta$, the following holds:

$$\sup_{f \in \mathcal{F}}|\tilde{\mathcal{L}}(f, S) - \mathcal{L}(f, P)| \le \sup_{f \in \mathcal{F}}\left(|\mathcal{L}(f; P) - \tilde{\mathcal{L}}(f, \tilde{P})| + |\tilde{\mathcal{L}}(f, S) - \tilde{\mathcal{L}}(f, \tilde{P})|\right)$$

$$(\text{Lemma } 20) \le \Pr_{(x,y)\sim P}[y \notin Y_k^x] + \sup_{f \in \mathcal{F}}|\tilde{\mathcal{L}}(f, S) - \tilde{\mathcal{L}}(f, \tilde{P})|$$

$$\le \Pr_{(x,y)\sim P}[y \notin Y_k^x] + 2 \cdot \mathcal{R}_{n,D}(\tilde{\ell} \circ \mathcal{F}) + \sqrt{\frac{\ln(2/\beta)}{n}}, \tag{24}$$

where inequality (24) follows from Lemma 21 with

$$\tilde{\ell} \circ \mathcal{F} := \left\{g : \mathcal{X} \times \mathcal{Y} \to [0, 1], \ g(x, y) = \tilde{\ell}(f(x), (x, y))|\ f \in \mathcal{F}\right\}.$$

Finally, we have that:

$$\mathcal{R}_{n,D}(\tilde{\ell} \circ \mathcal{F}) = \mathbb{E}_S\left[\mathbb{E}_{\sigma_1,\ldots,\sigma_n}\left[\sup_{f \in \mathcal{F}}\left(\frac{1}{n}\sum_{i=1}^{n}\sigma_i\tilde{\ell}(f(x_i), (x_i, y_i))\right)\right]\right]$$

$$\le \tilde{L} \cdot \mathbb{E}_S[\hat{\mathcal{R}}_{n,S_\mathcal{X}}(\mathcal{F})] \tag{25}$$

$$= \tilde{L} \cdot \mathcal{R}_{n,D_\mathcal{X}}(\mathcal{F}), \tag{26}$$

where (25) follows from Lemma 22 (with $\Phi_i$ set to the function $\tilde{\ell}(\cdot, (x_i, y_i))$ for all $i \in \{1, \ldots, n\}$, and with $S_\mathcal{X}$ denoting the projection of $S$ on $\mathcal{X}$), and from Lemma 19 with

$$\tilde{L} = L \cdot \frac{1 + k \cdot p_{k,\varepsilon}}{1 - k \cdot p_{k,\varepsilon}}. \tag{27}$$

Inequality (18) now follows by combining (24), (26), and (27). $\square$