# OpenReview forum: "Deep Learning with Label Differential Privacy"
_NeurIPS.cc/2021/Conference — NeurIPS 2021 Poster_

### Official Review · Reviewer_Jhzq · 2021-07-14

**Rating:** 6
**Confidence:** 3

**Summary:**

This study considers the label differential private training of the classification model with DNN.  The basic idea is the randomized response, while the labels with top-k prior probabilities are returned when labels are randomized. Furthermore, given a prior label distribution, the proposed scheme can choose k adaptively so that the probability that the input label is returned by the mechanism is maximized. The proposed mechanism is proved to be optimal in the sense that the probability that the input label is returned by the mechanism is equal or greater than any label-DP mechanism.

In application to model training, the authors proposed a multi-stage strategy. Splitting the training data into several subsets and starting from the uniform label prior distribution, it gradually updates the label prior used for the mechanism. The performance of the presented mechanism is demonstrated with CIFAR10 and movie lens dataset with comparison to DP-SGD and its variants. The results show that the classification performance of their label-DP mechanism is significantly better than the existing DNN model with differential privacy.







**Ethical Concerns:**

No.

**Limitations And Societal Impact:**

I could not find any paragraphs that mention the limitations and societal impact of this study.
Since the proposed scheme basically improves privacy, it would have a good social impact. However, differently from regular DP, the coverage of label DP is more restricted compared to regular DP. So it would be nice to explain in what situations label-DP is sufficient for privacy protection.

**Main Review:**

The story and contribution are clear. The reviewer would like to consider the following issues in revision:

(1) The proposed mechanism is label-DP while the comparison methods are regular DP. So direct comparison is not fair. The discussion section should be revised considering this point.

(2) In Theorem 5, the authors proved that the excess population loss is independent on the dimension of the parameter space, which would be greatly advantageous in the training of DNN. Is it possible to demonstrate this effect experimentally?

(3) In Collolary 6, a probability is contained in the complexity, which looks weird. I understand that this is an informal version of the theorem, but I think some remarks on this point should be contained in the main body to clarify why this term is contained in O().



**Time Spent Reviewing:**

5

---

> ### Author Response · Authors · 2021-08-10
> **Response**
>
> **(1) Comparison to regular DP.**
> It is indeed easier to protect the privacy of a label than that of features and labels—we will make this remark more explicitly in the text. We compare with DP-SGD (the most popular method in DP deep learning) since there are no other natural baselines under the label DP setting. We will clarify this in the revision.
>
> **(2) Theorem 5.**
> This theorem is under the convex setting, which does not directly apply to the non-convex neural network learning setting in our experiments. Interestingly, our empirical results show that we can successfully train full ResNet models with lots of parameters, while DP-SGD that protects both labels and inputs need to use simplified network architectures as the noise scale grows with the parameter dimension. This suggests that a similar phenomenon might also hold for the non-convex setting.
>
> **(3) Corollary 6.**
> Thank you for the comment. Indeed we briefly mention in the paragraph after Corollary 6 that this term, in some sense, represents the quality of the prior—the smaller it is the better the prior—and we pay the penalty in the excess population loss when our prior is inaccurate. We will make this clearer in the revision.
>
> **Limitation.**
> We will expand on the limitations of our work in the revision, and stress that we only protect the labels but not the inputs, which might not be a sufficient privacy protection in all settings.

---

### Official Review · Reviewer_cLnW · 2021-07-16

**Rating:** 8
**Confidence:** 3

**Summary:**

This paper studies a Randomized Response algorithm with prior (RRwithPrior), which takes as input a label y, a prior distribution p on the possible labels [K], and outputs a randomized label aiming to maximize the probability that the output is y, while preserving epsilon-DP. The first claim of the paper is that the proposed algorithm is optimal among all epsilon-DP algorithms, given that the label y is actually distributed according to the prior distribution p. This is proven by arguing that RRwithPrior gives the optimal solution to a Linear Program that models this problem.

Then, the authors use the RRwithPrior algorithm as a subroutine to LP-MST, a multi-stage algorithm for training NNs with label differential privacy (LabelDP). LabelDP is a weaker notion of privacy where neighboring data sets differ on the label of a single example. The algorithm proceeds in stages, in each of which a model is trained on disjoint partitions of the dataset and the private labels are protected using RRwithPrior. An important thing to note is that LP-MST does not require a prior and is essentially boosting the prior that RRwtihPrior uses: it starts with the uniform distribution over all [K] classes, then labels the first batch of examples according to RRwithPrior with this uniform prior and moves to the next stage using the new model as a prior. LP-MST with one stage (called LP-1ST) is equivalent to Randomized Response (without prior). The performance of the epsilon-LabelDP model output by variations of LP-MST is empirically evaluated against other methods, demonstrating increased accuracy. The authors also experiment with new advances in self-supervised learning to retrieve useful priors.


The above are the main contributions of the paper, but this work also includes a study of SGD with LabelDP. The authors give a version of SGD with Randomized Response which satisfies epsilon-LabelDP and achieves a excess true risk in the order of K/(epsilon*sqrt(n)), hiding the dependence on the diameter of the parameter space and the Lipschitz constant. They also give a (not very tightly matching) lower bound of 1/\sqrt(epsilon*n). They extend the study to (epsilon, delta)-LabelDP, saving a sqrt(K) factor from the upper bound. The lower bound above works for (epsilon, delta)-LabelDP as well so in this case the gap between the two is smaller. In the process of deriving the lower bound, the authors give lower bounds on the empirical loss and give an extension of a known reduction from private SCO to private ERM from Bassily et al. to the case of LabelDP. They finally consider a slightly different algorithm and give bounds that depend on the quality of the priors.


**Limitations And Societal Impact:**

I think that there is a point that could be stressed more: making the distinction between cases where features+labels vs just labels should be protected is crucial. I think that the (weaker) type of privacy that is guaranteed by these methods should be made clear to the practitioners who might use these methods for their good accuracy.

**Main Review:**

Strengths:
- RRwithPrior seems to be a simple primitive which could be useful in several settings (e.g. as a building block as in the current paper) and this paper gives a clean analysis of its guarantees.
- For cases when label DP is appropriate, this approach seems to give a significant empirical performance boost compared to the stricter DP guarantee, making it potentially much more useful in practice.
- Although theoretical results in the paper depend on the assumption that the labels are indeed drawn from the known prior p, experiments with different priors show that in practice the approach is more robust. In the case of SCO, the authors give bounds that depend on the quality of the prior.

Weakness:
- I found it a bit hard to extract a main story from the paper. But I think that this is partly due to the fact that it includes a lot of results in an effort to study LabelDP and NN from many perspectives.

*Score justification*:
I think that overall this paper aims to do an extensive study, both theoretical and experimental, of the perks of LabelDP for NN, and it manages to demonstrate its usefulness.

*Detailed comments/typos*:
- It might be more convincing to motivate the use of priors with examples, as is done for LabelDP in the introduction, and add these to the sentence “In many real world applications, a prior distribution about the labels could be publicly obtained from domain knowledge and help the learning process”.
- lines 183+184: I was not sure what the proof referred to, because I did not expect an observation to have a proof, so it might be useful to indicate it. Also in the first line of the proof, shouldn’t that be epsilon-LabelDP
- Theorem 9: right bracket missing



**Time Spent Reviewing:**

6

---

> ### Author Response · Authors · 2021-08-10
> **Response**
>
> **Main story.**
> Our main story is (i) priors can be gainfully exploited to improve the traditional RandomizedResponse algorithm and (ii) a particularly useful application of (i) is in deep learning with label privacy, where the priors can be generated in a multistage training process.  We apologize for the story not being clear; we will amend the writing in the revision.
>
> **Priors.**
> Thank you for your suggestion.  We will add the sentence to the text.
>
> **Typos.**
> Thanks for noticing the typos.  As you note, the Observation is immediate; we added the proof only for the sake of completeness.
>
> **Limitations.**
> This is a great point - thank you.  Yes, practitioners should be fully made aware of the “weaker” protection labelDP offers.  We will expand on this in the Broader Impacts section.

---

### Official Review · Reviewer_De6X · 2021-07-17

**Rating:** 5
**Confidence:** 4

**Summary:**

The paper introduces a Randomized response (RR) concept to provide more accurate results at the same level of privacy protection of LabelDP. The method can be incorporated into LP-MST and SSL frameworks to improve model performance. To achieve that, the authors propose RRWithPrior, which maximizes the probability that the output label is correct, and a novel LP-MST for training deep neural networks with LabelDP and RRWithPrior. They show formal proof of optimality of RRWithPrior and some open research questions.

**Limitations And Societal Impact:**

This work does not have a negative societal impact that needs to be addressed.

**Main Review:**

Originality: The work is a novel combination of well-known techniques. The LabelDP is built on k-ary RR [Kairouz et al., 2016]. The method has not been evaluated with other privacy protection for labels, such as using Gaussian noise [1] or Laplace noise [2], or Functional mechanism [3].
[1] Wang, D., & Xu, J. (2019, May). On sparse linear regression in the local differential privacy model. In International Conference on Machine Learning (pp. 6628-6637). PMLR.
[2] Phan, H., Thai, M. T., Hu, H., Jin, R., Sun, T., & Dou, D. (2020, November). Scalable differential privacy with certified robustness in adversarial learning. In International Conference on Machine Learning (pp. 7683-7694). PMLR.
[3] Zhang, J., Zhang, Z., Xiao, X., Yang, Y., & Winslett, M. (2012). Functional mechanism: regression analysis under differential privacy. arXiv preprint arXiv:1208.0219.

Quality: The submission is technically sound and well supported by theoretical analysis and experiments.
Does the LP-MST currently work with non-interactive settings in local DP models? In which setting it works currently: centralized setting or federated setting? It seems to me that this is applied to local models, which are under the federated setting. But in that case, how can authors partition the dataset S into subsets and learn the model M in Algorithm 3.

Clarity: It is well-organized and has enough information to reproduce the results.
The self-supervised learning (SSL) framework looks interesting but not well-motivated in the main body from the beginning.

Significance: The results are interesting, and others can use the ideas.

**Time Spent Reviewing:**

5

---

> ### Author Response · Authors · 2021-08-10
> **Response**
>
> **Other privacy-protection.**
> Please note that since we are in a classification setting, adding Laplace/Gaussian noise to labels is not feasible.  The work of Wang & Xu considers label privacy only for linear regression.  Neither the work of Phan et al. nor the work of Zhang et al. concerns label privacy, as far as we understand; their focus is on protecting *both* the features and the labels.  We will add these clarifications to the revision.
>
> **Setting.**
> Our algorithm works in the *interactive* local DP setting, which is a variant of the federated setting that allows multiple rounds of interaction.  This allows us to partition the data and learn the model in a stage-by-stage manner.
>
> We also wish to point out that the interactive local DP model is more *restrictive* than the commonly used central DP setting (which is the model used in e.g., Abadi et al.’s DP-SGD paper) As a result, our algorithm also works in the latter model.
>
> **Self-supervised learning.**
> Thanks for pointing this out. We will clarify it further in the revision.

---

### Official Review · Reviewer_CTDz · 2021-07-17

**Rating:** 4
**Confidence:** 3

**Summary:**

This paper proposes the Randomized Response with Prior algorithm and then uses it as a preprocessing step to learn neural networks with label differential privacy. It shows the RRWithPrior algorithm is optimal in privatizing a label. It also provides extensive experimental results to show that labelDP is easier than protecting both the inputs and labels. Finally, it gives theoretical analysis to understand this phenomenon to the stochastic conveys optimization setting.

**Limitations And Societal Impact:**

The paper provides several open questions in the conclusions section. I would suggest the authors to add more discussion on the limitations. I don't see any negative societal impact of this paper.

**Main Review:**

The paper is well-written and easy to follow. Somehow, I felt the technical contribution is not strong enough. I've listed my main comments below.

1. I think in most applications, we are more interested in protecting the features rather than the label. I would suggest the authors provide some real applications as motivating examples.

2. Intuitively, LabelDP is much easier than protecting the entire individual record, so the LP-MST with other DP methods are not comparable. It's worth showing the accuracy is much better, but it's far from surprising.

3. I found the claim regarding Theorem 3 is a bit overly strong. R is just any \eps-DP algorithm when just randomizing a single label. (Just curious, is it possible that the RRWithPrior is equivalent to the exponential mechanism with properly defined score functions?)

4. If K is large, will the running time be too long?



**Time Spent Reviewing:**

4

---

> ### Author Response · Authors · 2021-08-10
> **Response**
>
>
> **1.**
> As we mentioned in the paper, online advertising is an important application where protecting the labels is sufficient.  For example, a recent proposal from Microsoft (https://github.com/WICG/privacy-preserving-ads/blob/main/MaskedLARK.md, Section titled "Masked Gradient Model Training") for aggregate reporting API in advertising measurements (which is to be a privacy-safe replacement for the widely-used third-party cookies), clearly calls out the conversion label as the quantity to be protected.  A similar motivation has been listed in a few recent research papers (https://eprint.iacr.org/2021/835 and https://arxiv.org/abs/2106.03408).
>
> **2.**
> It is indeed easier to protect the privacy of a label than that of features and labels—we will make this remark more explicitly in the text. On the other hand, we compare with DP-SGD (the most popular method in DP deep learning) since there are no other natural baselines under the label DP setting. And we stress that while *relatively* high accuracy is not surprising, our achieved accuracy being *absolutely* much closer to the non-private baseline is non-obvious and is very important for real world applications, where accuracy could be the most important factor.
>
> **3.**
> Thanks for pointing this out.  We will reword Theorem 3 statement to clarify that the comparison is against all algorithms that randomize a single label, given the prior.  It is unlikely that we could view RRwithPrior as a natural special case of the exponential mechanism since the former sometimes outputs a label with probability 0, which would require infinite scores for the latter.
>
> **4.**
> The running time of our algorithm is quasi-linear in $K$ (to sort the prior).  In this regard, we wish to point out: (i) it is essentially optimal in multistage training, since $O(K)$ time will be required to write down the prior after each stage, (ii) for reasonable values of $K$, the running time will be dominated by training time, and (iii) the focus of the current version is for small to modest label spaces.  We will clarify these in the revision.
>
> **Limitations.**
> Thanks for the suggestion.  We will expand on the limitations of our work in the revision, and stress that we only protect the labels but not the inputs, which might not be a sufficient privacy protection in all settings.

---

### Decision · Program_Chairs · 2021-09-27

**Decision:**

Accept (Poster)

**Comment:**

The paper presents an algorithm for learning ML models under label DP. While label DP provides weaker protection than regular DP, the authors make a convincing case that it may be sufficient for certain applications. The paper demonstrates that cleverly exploiting label DP allows significant increase in model utility compared to learning under full DP.

The reviews are divided with two reviewers recommending acceptance and two recommending rejection. However, the main arguments for rejection seem to be based on dismissal of label DP as a valid topic of research without clear supporting evidence.

As a result, I tend to recommend acceptance as the paper seems to yield significant improvements for a problem with at least some practical relevance.

While the paper includes very extensive bibliography, it seems that the authors have missed at least the following previous papers and methods for learning under label DP: "Differentially Private Regression with Gaussian Processes" and "Differentially Private Regression and Classification with Sparse Gaussian Processes".